# Inference-Time Conformal Reasoning with Valid Factuality Control for Large Language Models

**Ting Wang** [* 1]  **Yuanjie Shi** [* 2]  **Yan Yan** [2]  **Huan Zhang** [1]

## Abstract

Large language models (LLMs) increasingly perform multi-step reasoning, where intermediate claims form implicit directed acyclic graphs whose node correctness is structurally conditioned on their ancestors. This makes factuality uncertainty structural, rather than a trivial accumulation of node-wise errors, and necessitates inference-time uncertainty quantification over the reasoning structure. While conformal prediction (CP) offers flexible user-specified factuality control, existing work remains post-hoc and cannot intervene during generation. To fill the gap between CP's flexibility and its post-hoc limitation, we propose an *Inference-Time Conformal Reasoning (ITCR)* framework that integrates CP directly into reasoning graph generation. ITCR learns a structure-level factuality uncertainty function that aggregates claim-level factuality signals over reasoning graphs without complex modeling assumptions. We then design the non-conformity score based on graph-level factuality uncertainty and calibrate the conformal threshold to decide when to stop generation. We theoretically show such generation is nested, yielding valid coverage guarantees for factuality control. Experiments over multiple datasets and coverage objectives demonstrate empirically valid coverage. In downstream reasoning tasks, inference-time calibrated graphs yield more accurate generation than post-hoc pruned graphs.

## 1. Introduction

Large language models (LLMs) increasingly perform multi-step reasoning, producing sequences of intermediate claims

---
[*]Equal contribution  [1]University of Illinois Urbana-Champaign  [2]School of EECS, Washington State University. Correspondence to: Yan Yan <yan.yan1@wsu.edu>, Huan Zhang <huan@huan-zhang.com>.

*Proceedings of the $43^{rd}$ International Conference on Machine Learning*, Seoul, South Korea. PMLR 306, 2026. Copyright 2026 by the author(s).

with explicit dependency relations (Wei et al., 2022; Xu et al., 2025; Besta et al., 2025). These dependencies induce an implicit directed acyclic graph (DAG) (Besta et al., 2025; Jiang et al., 2024; Lightman et al., 2024; Zhang et al., 2024), where each node represents a claim and edges capture logical or evidential support (Li et al., 2025). Crucially, the correctness of a node is not self-contained but conditioned on its ancestors: an upstream error can invalidate all downstream claims (Lei et al., 2025; Wang et al., 2023a).

This dependence materially changes factuality uncertainty in reasoning (Creswell & Shanahan, 2022). Rather than arising from a simple aggregation of independent node-level errors, uncertainty is defined over the structure of the reasoning itself, reflecting which subsets of claims are unreliable (Paul et al., 2024). As reasoning is generated incrementally under ancestor dependencies, post-hoc or node-wise uncertainty estimates are structurally misaligned with the valid factuality of the resulting graph (Creswell & Shanahan, 2022; Holtzman et al., 2020). Reliable control therefore, requires inference-time uncertainty quantification at the level of the reasoning structure (Yao et al., 2023; Madaan et al., 2023), where validity is assessed and regulated over coherent collections of claims instead of isolated nodes.

Recent work (Rubin-Toles et al., 2025) applies conformal prediction (CP) (Vovk et al., 2005; Angelopoulos & Bates, 2022), a model-agnostic and distribution-free framework with user-specified coverage guarantees (Fontana et al., 2023), to factuality control in LLM reasoning. This method provides guarantees over coherent subsets of claims, ensuring that all retained claims are jointly consistent with ground-truth knowledge with probability at least $1 - \alpha$. However, existing methods operate in a post-hoc manner: a complete multi-step response is first generated, after which CP is applied over candidate subsets and then enforces ancestor-closure constraints on the retained claims. As calibration is defined only over generated content, CP cannot influence the generation process itself. This motivates the central question of this paper: *can we design inference-time conformal control for factuality, rather than post-hoc calibration, while preserving coverage guarantees for LLM reasoning?*

To address this question, we propose *Inference-Time Conformal Reasoning (ITCR)*, which integrates conformal pre-

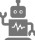 **Question:** Jordan wanted to surprise her mom with a homemade birthday cake. From reading the instructions, she knew it would take 20 minutes to make the cake batter and 30 minutes to bake the cake. The cake would require 2 hours to cool and an additional 10 minutes to frost the cake. If she plans to make the cake all on the same day, what is the latest time of day that Jordan can start making the cake to be ready to serve it at 5:00 pm?

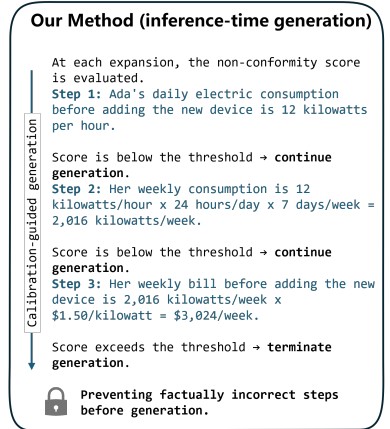

**Original Output with Ground Truth**

✔ **Step 1:** Ada's daily electric consumption before adding the new device is 12 kilowatts per hour.
✔ **Step 2:** Her weekly consumption is 12 kilowatts/hour x 24 hours/day x 7 days/week = 2,016 kilowatts/week.
✔ **Step 3:** Her weekly bill before adding the new device is 2,016 kilowatts/week x $1.50/kilowatt = $3,024/week.
✔ **Step 4:** After adding the new device, her daily consumption will be 12 kilowatts/hour + 2 kilowatts/hour = 14 kilowatts/hour.
✘ **Step 5:** Her weekly consumption will be 14 kilowatts/hour x 24 hours/day x 7 days/week = 3,024 kilowatts/week.
✘ **Step 6:** Her weekly bill after adding the new device is: 3,024 kilowatts/week x $1.50/kilowatt = $4,536/week.
✘ **Step 7:** The difference between her weekly bill before and after adding the new device is: $4,536/week - $3,024/week = $1,512/week.
✘ **Step 8:** Therefore, the difference between Ada's weekly electric bill before and after she adds the new device is $1,512.

**Existing Method (post-hoc)**

**Step 1:** Ada's daily electric consumption before adding the new device is 12 kilowatts per hour.
**Step 2:** Her weekly consumption is 12 kilowatts/hour x 24 hours/day x 7 days/week = 2,016 kilowatts/week.
[Score is larger than threshold, **pruning**.]

*Lacking ancestor, pruning*

**Step 3:** Her weekly bill before adding the new device is 2,016 kilowatts/week x $1.50/kilowatt = $3,024/week.
**Step 4:** After adding the new device, her daily consumption will be 12 kilowatts/hour + 2 kilowatts/hour = 14 kilowatts/hour.
**Step 5:** Her weekly consumption will be 14 kilowatts/hour x 24 hours/day x 7 days/week = 3,024 kilowatts/week.
**Step 6:** Her weekly bill after adding the new device is: 3,024 kilowatts/week x $1.50/kilowatt = $4,536/week.
**Step 7:** The difference between her weekly bill before and after adding the new device is: $4,536/week - $3,024/week = $1,512/week.
**Step 8:** Therefore, the difference between Ada's weekly electric bill before and after she adds the new device is $1,512.
[Lacking ancestor step, **pruning**.]

**Our Method (inference-time generation)**

*Calibration-guided generation*

At each expansion, the non-conformity score is evaluated.
**Step 1:** Ada's daily electric consumption before adding the new device is 12 kilowatts per hour.

Score is below the threshold → **continue generation.**
**Step 2:** Her weekly consumption is 12 kilowatts/hour x 24 hours/day x 7 days/week = 2,016 kilowatts/week.

Score is below the threshold → **continue generation.**
**Step 3:** Her weekly bill before adding the new device is 2,016 kilowatts/week x $1.50/kilowatt = $3,024/week.

Score exceeds the threshold → **terminate generation.**

🔒 **Preventing factually incorrect steps before generation.**

*Figure 1.* **Prior post-hoc pruning (Rubin-Toles et al., 2025) vs. our inference-time generation** on an example from LLaMA-3.1-8B-Instruct with GSM8K dataset (Cobbe et al., 2021). **(Left)** original output with ground truth. Intermediate steps are annotated as correct (blue) and incorrect (red). **(Center)** existing post-hoc method. After full generation, conformal filtering prunes steps exceeding the threshold, followed by a second pass removing steps with missing ancestors. **(Right)** our method. Conformal calibration is performed at inference time: the score is checked at each expansion, and generation stops immediately when the threshold is crossed. Center and right panels use the same confidence score and 85% confidence level; both outputs are guaranteed to contain no factually incorrect steps.

diction directly into the generation of reasoning graphs. Intuitively, ITCR performs factuality control at inference time, using calibrated uncertainty to decide whether the reasoning graph should expand or terminate. Enabling such control requires calibration over partially generated reasoning graphs. Accordingly, ITCR operates on uncertainty defined directly over intermediate subgraphs. Figure 1 illustrates this distinction by contrasting prior post-hoc pruning with the inference-time generation paradigm of ITCR.

Specifically, ITCR learns a structure-level factuality uncertainty function that maps an ancestor-closed reasoning subgraph to a scalar uncertainty score by aggregating inference-time claim-level signals. Based on this uncertainty function, we define a graph-level non-conformity score and calibrate a conformal threshold on held-out data. At test time, reasoning proceeds by incrementally expanding the graph and terminates once the non-conformity score exceeds the calibrated threshold, returning the current ancestor-closed subgraph as the final output. We show that this procedure induces a nested sequence of reasoning subgraphs, enabling valid coverage guarantees for factuality control via conformal prediction. Across multiple datasets and coverage targets, ITCR achieves empirically valid coverage. Moreover, inference-time calibrated reasoning graphs consistently yield more accurate generations than post-hoc pruned graphs in downstream reasoning tasks.

**Contributions.** The key contributions of this paper include:

- We propose Inference-Time Conformal Reasoning

(ITCR), which integrates conformal prediction into reasoning graph generation to enable inference-time factuality control rather than post-hoc in prior work.

- We show that ITCR induces nested reasoning graphs, yielding valid conformal coverage guarantees over structured reasoning outputs.

- Experiments across multiple reasoning benchmarks and LLM backbones show that ITCR achieves empirically valid coverage and improves reasoning accuracy by 18.77% on average.

## 2. Problem Setup and Motivation

**Notations.** Let $\mathcal{X}$ denote the input space (e.g., questions or prompts), and let $X \in \mathcal{X}$ be a given input instance. Let $P_X$ denote the data distribution over inputs $X \in \mathcal{X}$. Denote $\mathcal{C}$ as the claim space, i.e., the set of atomic factual statements. For structured reasoning outputs, an input $X$ induces a directed acyclic graph (DAG) $G_X = (V, E)$, where each node $v \in V$ corresponds to a claim $c_v \in \mathcal{C}$, and each directed edge $(u, v) \in E$ indicates that claim $c_v$ depends on claim $c_u$. When the dependence on $X$ is clear from context, we write $G$ for simplicity. We assume $G$ is acyclic but make no further assumptions on its structure. For any node $v \in V$, we define its ancestor set as $\text{Anc}(v) := \{u \in V : u \rightsquigarrow v \text{ in } G\}$, where $u \rightsquigarrow v$ denotes the existence of a directed path from $u$ to $v$. A reasoning subgraph is denoted by $U = (V_U, E_U) \subseteq G$, where $V_U \subseteq V$ and $E_U \subseteq E$. Such a subgraph $U$ is ancestor-closed if

$\forall v \in V_U, \mathrm{Anc}(v) \subseteq V_U$. Let $\mathcal{T} \subseteq V$ denote the set of factually correct nodes in the reasoning graph. For each node $v \in V$, we define a node-level factuality uncertainty score $\mathrm{fu}(v) \in [0, 1]$, which measures the likelihood that claim $c_v$ violates factual correctness given its supporting ancestors.

**Conformal Language Model Reasoning.** Existing conformal reasoning approaches (Rubin-Toles et al., 2025) perform factuality control on a fixed reasoning graph $G$ using node-level factuality uncertainty scores $\mathrm{fu}(v)$. For a given threshold $\tau$, a candidate subgraph is constructed through a *two-stage pruning procedure*. First, nodes are filtered by score thresholding, retaining the set $V_\tau = \{v \in V : \mathrm{fu}(v) \leq \tau\}$. Second, ancestor closure is enforced by iteratively removing any node $v \in V_\tau$ such that $\mathrm{Anc}(v) \nsubseteq V_\tau$, yielding an ancestor-closed subgraph $U_\tau \subseteq G$. Conformal prediction (CP) is then carried out over the threshold-indexed family $\{U_\tau\}$, and the calibrated output corresponds to selecting a subgraph that satisfies the coherent factuality guarantee at level $1 - \alpha$. Accordingly, factuality control in this setting is defined purely as a selection problem over a fixed graph, rather than as a control mechanism during generation.

To address this limitation, we formulate an inference-time objective that directly outputs an ancestor-closed subgraph $\widehat{U} = (V_{\widehat{U}}, E_{\widehat{U}})$ during generation. Our goal is to control the factuality of $\widehat{U}$ with two different coverage targets.

**No-False Coverage.** The no-false target requires that the predicted reasoning subgraph contains no factually incorrect nodes. The corresponding coverage guarantee is:

$$\mathbb{P}\left(V_{\widehat{U}} \subseteq \mathcal{T}\right) \geq 1 - \alpha. \tag{1}$$

Among all ancestor-closed subgraphs satisfying this guarantee, the target output is maximal with respect to inclusion.

**No-Miss Coverage.** The no-miss target requires that all factually correct nodes are included in the predicted subgraph. Formally, the corresponding coverage guarantee is:

$$\mathbb{P}\left(\mathcal{T} \subseteq V_{\widehat{U}}\right) \geq 1 - \alpha. \tag{2}$$

Accordingly, the target output is the minimal ancestor-closed subgraph satisfying the coverage guarantee.

## 3. Inference-Time Subgraph Prediction

### 3.1. Learning Graph-Level Factuality Uncertainty

**Why Graph-Level Uncertainty Is Necessary for Reasoning.** Inference-time factuality in structured reasoning is inherently a graph-level property: The correctness of a claim depends on the presence and correctness of its supporting ancestors, and valid outputs must therefore be ancestor-closed (Madaan et al., 2023). This structural dependence makes factuality uncertainty conditional across nodes, so that node-wise estimates alone cannot support inference-time control

and are limited to post-hoc pruning or structural repair after a full reasoning graph is generated.

This limitation is reflected in prior conformal approaches (Rubin-Toles et al., 2025), which enforce factuality guarantees only after complete graph generation via subgraph selection and repair, revealing a mismatch between local uncertainty and graph-level reliability objectives. To enable principled inference-time generation under ancestor constraints, we thus require a notion of graph-level factuality uncertainty that quantifies the risk of a reasoning subgraph.

However, there is no general principle for aggregating node-wise uncertainty into a structurally consistent, conformal-calibratable graph-level measure (Wang et al., 2024a). We therefore propose to learn such a graph-level factuality uncertainty function from data, using conformal supervision to obtain valid coverage guarantees during generation.

**Binary Supervision and Black-Box Uncertainty Modeling.** Concretely, we parameterize a subgraph-level factuality uncertainty function:

$$\mathrm{FU}_\theta : (U, \{\mathrm{fu}(v)\}_{v \in V_U}) \mapsto \mathbb{R}_+, \tag{3}$$

where $\mathrm{FU}_\theta(U, \{\mathrm{fu}(v)\}_{v \in V_U})$ evaluates the factuality uncertainty of the currently generated subgraph $U$. $\mathrm{FU}_\theta(U, \{\mathrm{fu}(v)\}_{v \in V_U})$ is intended to reflect uncertainty of the factual violation of a subgraph $U$.

We construct supervision at the subgraph level using factual violations. Graph-level factuality is a binary predicate, where each subgraph $U$ is either consistent with the reference world or not (Creswell & Shanahan, 2022). Therefore, learning $\mathrm{FU}_\theta$ reduces to binary classification, with its continuous output interpreted as a factuality uncertainty score for ranking. In principle, $\mathrm{FU}_\theta$ can be instantiated by any permutation-invariant model over sets or graphs, including linear models (Wang et al., 2018), tree-based models (Shi et al., 2023), or neural networks (Wu et al., 2020).

Importantly, the accuracy of the learned factuality uncertainty function does not affect the validity of the conformal coverage guarantees. CP treats the learned factuality uncertainty score as a black-box quantity and provides coverage regardless of how it is obtained. Learning primarily influences efficiency, rather than coverage itself.

### 3.2. Conformal Calibration and Inference Algorithm

**Nested property for inference-time generation.** Previous work (Shafer & Vovk, 2008; Gupta et al., 2022) shows that standard conformal prediction (CP) can be formulated via a family of nested prediction sets $\{F_t\}_{t \in \mathcal{T}}$, where calibration selects the smallest index $t_\alpha$ to achieve valid coverage. In this view, the non-conformity score is the inverse map $r(x, y) = \inf\{t : y \in F_t(x)\}$, and thresholding is meaningful because the sets are nested: $F_{t_1} \subseteq F_{t_2}$ for $t_1 \leq t_2$.

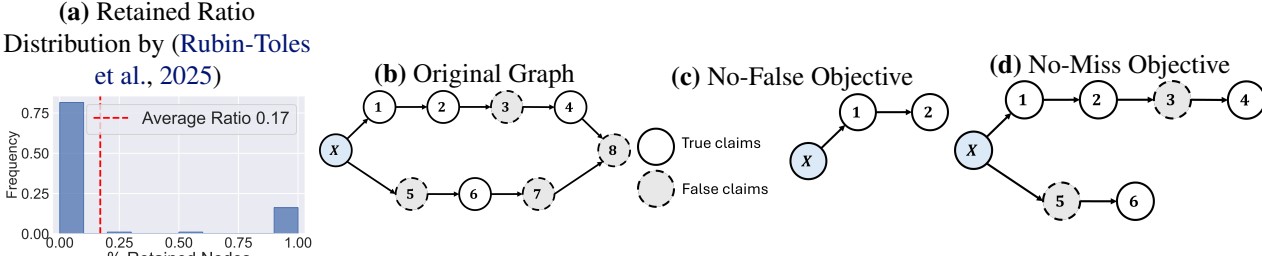

*Figure 2.* **Motivation for the No-Miss Objective. (a)** Distribution of retained node ratios produced by previous method (Rubin-Toles et al., 2025) on the QA dataset (chen et al., 2023). Although the average retained ratio is moderate (0.17), a substantial fraction of samples collapse to 0 retained nodes, corresponding to complete abstention. **(b)** An example of an original reasoning graph containing both true (solid) and false (dotted) claims. **(c)** No-false (precision-oriented) objective, which outputs a maximal ancestor-closed subgraph containing no false claims and may yield overly small outputs or complete abstention. **(d)** No-miss (recall-oriented) objective, which outputs the minimal ancestor-closed subgraph containing all true claims, at the cost of allowing a limited number of uncertain nodes.

---

**Algorithm 1** Inference-Time Conformal Subgraph Generation via Threshold Stopping

---

1: **Input:** Test instance $X_{n+1}$; non-conformity score $S(\cdot)$; threshold $\tau_\alpha$.
2: Initialize $U^0$ (e.g., root-only subgraph) and set $t \leftarrow 0$
3: **while** true **do**
4:     **if** $S(U^t) \leq \tau_\alpha$ **then**
5:         $\widehat{U} \leftarrow U^t$
6:         Obtain the next expanded subgraph $U^{t+1}$ from generator
7:         $t \leftarrow t + 1$
8:     **else**
9:         **break**
10:     **end if**
11: **end while**
12: **return** $\widehat{U}$

---

Inference-time reasoning over graphs induces an analogous but sequential structure: the output is constructed as an inclusion-ordered chain of ancestor-closed subgraphs $U^1 \subset U^2 \subset \cdots \subset U^{T_G}$, where $\exists T_G < +\infty$, s.t. $U^{T_G} = G$. The calibrated threshold $\tau$ is queried repeatedly along this chain to decide when to stop. To transfer the nested-set view of CP to this setting, the non-conformity score must be monotone under subgraph expansion.

**Definition 3.1** (Nested property for non-conformity score during generation). A non-conformity score $S$ satisfies the nested property if it is monotone under subgraph inclusion, i.e., $S(U^1) \leq S(U^2) \leq \cdots \leq S(U^{T_G})$ for any $G$.

*Remark* 3.2. Inference terminates at the first index $t$ such that $S(U^t) > \tau$. The nested property makes this violation irreversible: for all $t' > t$, $S(U^{t'}) \geq S(U^t) > \tau$, so generation cannot re-enter the admissible region. Without nestedness, one may have $S(U^t) > \tau$ but $S(U^{t+1}) < \tau$, which invalidates calibrated stopping.

The nested property is a structural requirement specific to

sequential generation and is not guaranteed by a learned factuality predictor alone. Although $\mathrm{FU}_\theta(U, \{\mathrm{fu}(v)\}_{v \in U})$ estimates graph-level uncertainty, its sigmoid-transformed output is unconstrained and need not be monotone along $\{U^t\}_{t=0}^{T_G}$. Therefore, unlike standard CP, where any black-box score can be post-hoc calibrated, inference-time control requires a non-conformity score whose form enforces nestedness, so that the calibrated threshold retains a consistent interpretation as a stopping criterion.

To this end, we define

$$S(U) = 1 - \sigma\big(\mathrm{FU}_\theta(U, \{\mathrm{fu}(v)\}_{v \in V_U})\big) + \lambda|V_U|, \quad (4)$$

where $(\cdot)^+ := \max(\cdot, 0)$, $\sigma(\cdot)$ is the sigmoid and $\lambda \in (0, 1)$. The first term captures model-based factuality uncertainty, while the size penalty enforces monotonicity by construction, reflecting that uncertainty accumulates under subgraph expansion. Coverage validity is ensured by conformal calibration for any fixed $\lambda$ (Vovk et al., 2005; Papadopoulos, 2008). Meanwhile, $\lambda$ serves as an efficiency hyperparameter (Angelopoulos et al., 2021; Huang et al., 2024).

Coverage holds for any fixed $\lambda$, while nested monotonicity imposes a structural requirement on $\lambda$ to be sufficiently large to offset non-monotone fluctuations of the learned term. The following proposition gives a sufficient condition on $\lambda$.

**Proposition 3.3** (Condition for nested monotonicity). *Let* $B(U) = 1 - \sigma(\mathrm{FU}_\theta(U, \{\mathrm{fu}(v)\}_{v \in V_U}))$. *For each input* $X \sim P_X$, *consider all ancestor-closed expansions* $U \subset U' \subseteq G_X$, *and define*

$$\kappa(X) := \sup_{U \subset U' \subseteq G_X} \frac{(B(U) - B(U'))^+}{|V_{U'}| - |V_U|}, \quad \kappa := \sup_{X \sim P_X} \kappa(X).$$

*If* $\lambda \geq \kappa$, *then* $S(U)$ *satisfies the nested property.*

*Remark* 3.4. The constant $\kappa$ characterizes the worst-case non-monotone fluctuation of the learned term $B(U)$ per added node along inference-time expansions. When $\lambda \geq$

$\kappa$, the size penalty dominates these fluctuations, ensuring that $S(U)$ increases under any ancestor-closed expansion and that threshold crossing is irreversible. In practice, the global constant $\kappa$ is defined via a supremum over all inputs and expansions and is intractable to compute. Instead, we examine the instance-wise quantity $\kappa(X)$ along calibration trajectories to characterize the scale of non-monotonicity induced by the learned score. The empirical distribution of $\kappa(X)$ (see Fig. 4) provides a reference for initializing $\lambda$ and for exploring the efficiency–robustness tradeoff in practice.

**Calibration and Inference Algorithms.** Given the non-conformity score $S(U)$ in (4) and for each calibration graph $G$, we extract its ground-truth-consistent subgraph and compute the corresponding score $\{S(U^t)\}_i^{T_G}$, keeping the learned mapping and all hyperparameters fixed. We assume that the calibration graphs and the test graph are i.i.d. samples from the same underlying distribution (Liao et al., 2021; Veličković et al., 2018), and therefore exchangeable as required for conformal calibration (Shafer & Vovk, 2008; Vovk et al., 2005). For a target miscoverage level $\alpha \in (0, 1)$, we set the inference threshold $\tau$ according to the corresponding coverage targets, detailed in Section 3.3 and 3.4. This choice guarantees marginal coverage $1 - \alpha$ and defines the stopping criterion for inference-time generation.

At test time, inference proceeds by incrementally constructing a predicted reasoning subgraph according to a fixed expansion rule, summarized in Algorithm 1. At each step $t$, generation continues if $S(U^t) \leq \tau_\alpha$, in which case the current subgraph is accepted and the next subgraph $U^{t+1}$ is obtained from the generator (Lines 3–7). Once the threshold is exceeded, i.e., $S(U^t) > \tau_\alpha$, the procedure terminates and returns the last accepted subgraph $\widehat{U}$ as the final prediction (Lines 9–12). By monotonicity of the non-conformity score, threshold crossing is irreversible, guaranteeing that the procedure requires no backtracking or post-hoc pruning.

### 3.3. No-False Coverage

We first consider the precision-oriented objective, no-false coverage (Eq (1)). A failure occurs if the predicted subgraph contains at least one false node. Equivalently, no-false coverage amounts to controlling the false-positive rate (FPR) over subgraphs that contain false nodes. This reduces calibration to a one-sided conformal problem on a restricted subset, rather than over all candidate subgraphs.

During calibration, for each input $X_i$ that admits a false node, we select the minimal ancestor-closed subgraph $U_i^{\mathrm{nf}}$ such that $V_{U_i^{\mathrm{nf}}} \nsubseteq \mathcal{T}$, and set the no-false threshold $\tau_\alpha^{\mathrm{nf}}$ as the empirical $\alpha$-quantile of the scores $\{S(U_i^{\mathrm{nf}})\}$. Under nested generation and score monotonicity, accepting any later bad subgraph implies accepting this earliest one, so calibrating on these scores suffices for no-false coverage.

**Theorem 3.5** (No-false coverage guarantee). *Suppose $S(\cdot)$ satisfies the nested property. Let $\widehat{U}^{nf}$ denote the output of Algorithm 1 with threshold $\tau_\alpha^{nf}$. Then the inference procedure achieves no-false coverage at level $1 - \alpha$, i.e., $\mathbb{P}(V_{\widehat{U}^{nf}} \subseteq \mathcal{T}) \geq 1 - \alpha$.*

*Remark* 3.6. The coverage guarantee in the above theorem is model-agnostic and distribution-free: it relies only on conformal calibration and the nested property of the nonconformity score, not on the learning performance of $\mathrm{FU}_\theta$. Improving $\mathrm{FU}_\theta$ mainly affects efficiency (e.g., retaining larger subgraphs or terminating later), while the validity guarantee remains unchanged. Since the same score definition is applied to both calibration and test inputs, exchangeability is preserved, so the conformal guarantee holds.

### 3.4. No-Miss Coverage

We next consider a recall-oriented reliability objective, referred to as no-miss coverage (Eq (2)). As illustrated in Fig. 2, no-false coverage enforces a stringent precision criterion that excludes any reasoning subgraph containing false claims. In structured reasoning graphs, where true and false nodes may be interleaved, this requirement can be overly restrictive, leading to truncated or degenerate outputs. The no-miss coverage adopts a complementary, recall-oriented notion of reliability by requiring inclusion of all factually correct nodes, while permitting limited uncertainty when necessary. This relaxation enables more complete reasoning subgraphs and alleviates excessive abstention.

For each calibration example, we consider the minimal ancestor-closed subgraph $U^{\mathrm{nm}}$ that contains all true nodes. The no-miss threshold $\tau_{1-\alpha}^{\mathrm{nm}}$ is selected as the $(1 - \alpha)$-quantile of $\{S(U^{\mathrm{nm}})\}$ over these minimal subgraphs.

**Theorem 3.7** (No-miss coverage guarantee). *Suppose that $S(\cdot)$ satisfies the nested property. Let $\widehat{U}^{nm}$ be the output returned by Algorithm 1 with threshold $\tau_{1-\alpha}^{nm}$. Then the inference procedure satisfies no-miss coverage at level $1-\alpha$, i.e., $\mathbb{P}(\mathcal{T} \subseteq V_{\widehat{U}^{nm}}) \geq 1 - \alpha$.*

*Remark* 3.8. The no-miss coverage guarantee in Theorem 3.7 follows from conformal calibration and is independent of the accuracy of $\mathrm{FU}_\theta$. The learned function $\mathrm{FU}_\theta$ influences efficiency only: higher accuracy yields more aggressive and precise pruning, removing more false nodes while retaining all true ones under the same coverage rate.

## 4. Experiments

### 4.1. Datasets

We evaluate on two standard mathematical reasoning benchmarks, MATH (Hendrycks et al., 2021) and GSM8K dataset (Cobbe et al., 2021). To evaluate cross-domain generalization, we additionally include a world-knowledge question answering (QA) benchmark (chen et al., 2023). We

*Table 1.* **Empirical coverage and efficiency under the *no-miss* and *no-false* coverage targets on MATH, GSM8K, and QA benchmarks** with different $\alpha \in \{0.05, 0.1\}$ values. For each coverage target and miscoverage level $\alpha$, we report the empirical coverage (targeting $1 - \alpha$) and the corresponding efficiency (higher is better). ✓ indicates that the empirical coverage satisfies the target level $1 - \alpha$ within a tolerance of 0.01, while × indicates violation of the coverage requirement. For methods that fail to satisfy the coverage requirement (×), the corresponding efficiency values (shown in gray) are reported for completeness but are not considered in efficiency comparisons. Results are reported as mean $\pm$ standard deviation over 100 independent runs. It is clear that ITCR consistently achieves valid coverage with the best efficiency across all datasets, coverage targets, and miscoverage levels.

| Dataset | Method | No-miss | | | | No-false | | | |
| --- | --- | --- | --- | --- | --- | --- | --- | --- | --- |
| | | $\alpha = 0.05$ | | $\alpha = 0.10$ | | $\alpha = 0.05$ | | $\alpha = 0.10$ | |
| | | $\text{Cov}^{\text{nm}}$ | $\text{Eff}^{\text{nm}}(\%)$ | $\text{Cov}^{\text{nm}}$ | $\text{Eff}^{\text{nm}}(\%)$ | $\text{Cov}^{\text{nf}}$ | $\text{Eff}^{\text{nf}}(\%)$ | $\text{Cov}^{\text{nf}}$ | $\text{Eff}^{\text{nf}}(\%)$ |
| MATH | CPL | × 0.144±0.19 | 63.72±0.15 | × 0.405±0.27 | 43.36±0.20 | × 0.548±0.07 | 37.04±0.16 | × 0.624±0.10 | 57.27±0.17 |
| | ITCR-SUM | × 0.903±0.20 | 12.58±0.21 | × 0.729±0.37 | 32.33±0.38 | ✓ 1.000±0.00 | 0.00±0.00 | ✓ 0.991±0.07 | 1.83±0.13 |
| | ITCR-AVG | × 0.917±0.08 | 12.93±0.08 | × 0.858±0.15 | 17.13±0.12 | ✓ **0.953**±0.06 | 16.50±0.19 | ✓ 0.945±0.07 | 22.80±0.23 |
| | ITCR-MAX | ✓ 0.942±0.08 | **11.73**±0.06 | × 0.878±0.13 | 17.15±0.09 | × 0.938±0.07 | 32.91±0.31 | ✓ 0.942±0.07 | **35.72**±0.32 |
| | **ITCR** | ✓ **0.947**±0.08 | 3.44±0.03 | ✓ **0.894**±0.10 | **6.64**±0.05 | ✓ 0.943±0.08 | **21.66**±0.23 | ✓ **0.908**±0.07 | 34.77±0.16 |
| GSM8K | CPL | × 0.169±0.05 | 80.35±0.07 | × 0.263±0.08 | 61.75±0.13 | × 0.853±0.06 | 18.40±0.06 | × 0.700±0.09 | 37.49±0.12 |
| | ITCR-MAX | × 0.938±0.04 | 6.50±0.03 | × 0.883±0.05 | 11.20±0.04 | ✓ 0.981±0.02 | 11.01±0.09 | ✓ 0.949±0.04 | 25.05±0.12 |
| | ITCR-SUM | × 0.931±0.04 | 5.24±0.04 | × 0.864±0.06 | 10.76±0.05 | ✓ 1.000±0.00 | 1.00±0.01 | ✓ 0.997±0.01 | 2.73±0.03 |
| | ITCR-AVG | × 0.917±0.05 | 7.10±0.04 | × 0.852±0.06 | 12.24±0.05 | ✓ 0.981±0.02 | 11.01±0.09 | ✓ 0.957±0.03 | 17.25±0.07 |
| | **ITCR** | ✓ **0.945**±0.04 | **2.15**±0.02 | ✓ **0.902**±0.06 | **4.13**±0.01 | ✓ **0.953**±0.03 | **18.06**±0.06 | ✓ **0.921**±0.04 | **26.94**±0.08 |
| QA | CPL | × 0.383±0.04 | 92.72±0.05 | × 0.442±0.05 | 78.60±0.08 | ✓ 0.947±0.04 | 7.73±0.05 | × 0.870±0.05 | 19.80±0.07 |
| | ITCR-MAX | ✓ **0.940**±0.04 | 5.29±0.04 | ✓ 0.902±0.05 | 9.67±0.04 | ✓ 0.943±0.04 | 10.26±0.06 | ✓ 0.911±0.05 | 18.36±0.08 |
| | ITCR-SUM | × 0.936±0.05 | 7.00±0.07 | × 0.859±0.07 | 19.55±0.09 | ✓ 0.970±0.02 | 3.66±0.02 | ✓ 0.969±0.02 | 4.92±0.02 |
| | ITCR-AVG | × 0.923±0.06 | 9.55±0.07 | × 0.855±0.07 | 19.51±0.09 | ✓ 0.955±0.04 | 7.39±0.06 | ✓ 0.925±0.04 | 16.34±0.07 |
| | **ITCR** | ✓ **0.957**±0.03 | **6.25**±0.02 | ✓ **0.900**±0.05 | **10.29**±0.03 | ✓ **0.948**±0.04 | **13.90**±0.07 | ✓ **0.899**±0.05 | **25.34**±0.08 |

leverage existing datasets from (chen et al., 2023; Mohri & Hashimoto, 2024), where individual subclaims are annotated for factual correctness, from which we derive subset-level coherent labels according to our ancestor-closure criterion. The dataset is randomly split into three disjoint subsets for learning the graph-level factuality uncertainty function, conformal calibration, and testing. Additional dataset details are provided in Appendix B. [1]

### 4.2. Conformal Prediction Performance

**Evaluation Protocol: CP metrics.** We evaluate the empirical coverage rate and the corresponding efficiency for the two different coverage targets: no-false coverage (defined in eq (1)) and no-miss coverage (defined in eq (2)). Let $D_{\text{te}}$ denote the test set. The specific metrics are summarized:

(i) **No-false coverage**: The corresponding empirical coverage rate is defined as $\text{Cov}^{\text{nf}} := \frac{1}{|D_{\text{te}}|} \sum_{i \in D_{\text{te}}} \mathbb{1}\left[ V_{\widehat{U}_i} \subseteq \mathcal{T}_i \right]$, where $\mathcal{T}_i$ denotes the set of factually correct nodes in the reasoning graph $G_i$ associated with test instance $i$. Under the same coverage type, the corresponding efficiency is measured by the average fraction of nodes *retained* in the predicted subgraph as: $\text{Eff}^{\text{nf}} := \frac{1}{|D_{\text{te}}|} \sum_{i \in D_{\text{te}}} \frac{\left| V_{\widehat{U}_i} \right|}{\left| V_{U_i} \right|}$, where higher $\text{Eff}^{\text{nf}}$ values indicate that a larger fraction of nodes in the reasoning graph is *retained* under the no-false constraint.

(ii) **No-miss coverage**: Its empirical coverage rate is defined

as: $\text{Cov}^{\text{nm}} := \frac{1}{|D_{\text{te}}|} \sum_{i \in D_{\text{te}}} \mathbb{1}\left[ \mathcal{T}_i \subseteq V_{\widehat{U}_i} \right]$. Under the same coverage type, the corresponding efficiency is measured by the average fraction of nodes *removed* in the predicted subgraph as: $\text{Eff}^{\text{nm}} := \frac{1}{|D_{\text{te}}|} \sum_{i \in D_{\text{te}}} \frac{\left| V_{U_i} \backslash V_{\widehat{U}_i} \right|}{\left| V_{U_i} \right|}$, where higher $\text{Eff}^{\text{nm}}$ values indicate that a larger fraction of nodes in the reasoning graph is *removed* under the no-miss constraint.

**Evaluation Protocol: CP Baselines.** We consider inference-time CP methods that operate during generation, as post-hoc pruning and inference-time generation correspond to different evaluation settings. Specifically, we include: (i) **CPL** (Mohri & Hashimoto, 2024), which estimates factuality risk at the individual-claim level using LLMs. To isolate the contribution of our learned graph-level factuality uncertainty function, we further evaluate several variants of our methods that only replace our learned function with heuristic aggregation rules over node-level factuality uncertainty scores: (ii) **ITCR-MAX**, which uses the maximum node-level factuality uncertainty scores as the graph-level score; (iii) **ITCR-SUM**, which uses the sum of node-level factuality uncertainty scores; (iv) **ITCR-AVG**, which uses the average node-level factuality uncertainty scores. In all experiments, $\lambda$ is initialized at a high empirical quantile of $\kappa(X)$ estimated from the corresponding calibration data. All methods are evaluated under the same experiment settings and conducted on a server equipped with NVIDIA A100 GPUs. We repeat experiments for all methods over 100 independent runs, with randomness arising from different splits of the calibration and test sets.

---

[1]Code: https://github.com/tinattw/ITCR

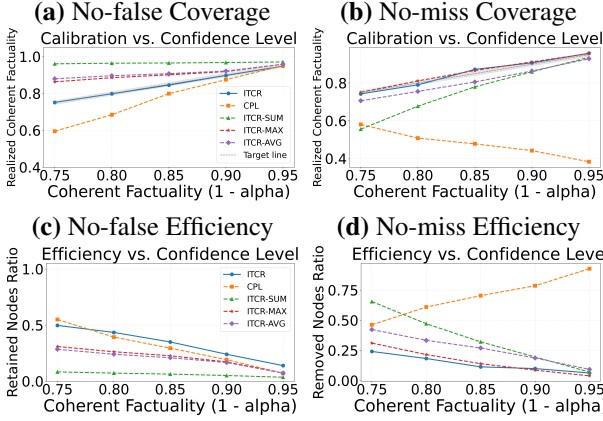

*Figure 3.* **Coverage and efficiency across target confidence levels** $1 - \alpha$ **under no-false and no-miss objectives** on QA dataset. Subfigure **(a)** and **(b)** report the achieved empirical coverage as a function of the target coverage, with the dashed gray line indicating the ideal calibration line. Subfigure **(c)** and **(d)** report the corresponding efficiency, measured by the retained node ratio, where lower values indicate more compact reasoning subgraphs. ITCR remains close to the target calibration line and exhibits higher efficiency with valid coverage than baselines across target coverage.

**Results: ITCR consistently achieves valid coverage with the best efficiency across all datasets, coverage targets, and miscoverage levels.** Table 1 summarizes the empirical coverage and efficiency of ITCR and its variants under both no-miss and no-false objectives. Across all settings, ITCR satisfies the target coverage guarantees, while its variants based on simple graph-level aggregations (MAX, SUM, AVG) achieve competitive performance but exhibit less stable validity. This highlights the benefit of learning a principled graph-level factuality uncertainty function rather than relying on heuristic aggregation. Among all methods that satisfy coverage, ITCR attains empirical coverage closest to the target level $1 - \alpha$ while achieving the highest efficiency, demonstrating a favorable trade-off between coverage guarantee and inference-time efficiency.

**Results: ITCR consistently achieves valid coverage with near-tight efficiency across target coverage levels.** Fig. 3 examines the coverage-efficiency trade-offs as the target coverage level $1 - \alpha$ varies from 0.75 to 0.95 with a range 0.05. Across all target coverage levels, ITCR consistently satisfies both no-miss and no-false coverage guarantees, while its empirical coverage closely tracks the target coverage $1 - \alpha$. At the same time, ITCR achieves the highest efficiency among all valid methods, with efficiency degrading smoothly as target coverage increases, reflecting near-tight inference-time generation rather than conservative over-coverage.

**Results: ITCR is invariant to the choice of uncertainty function class for coverage.** Throughout our experiments, we use a multilayer perceptron (MLP) to instantiate the graph-level factuality uncertainty function (defined in Eq

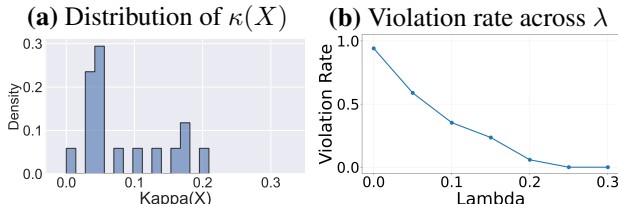

*Figure 4.* **Empirical validation of the nestedness condition in Proposition 3.3** on MATH dataset. **(a)** Distribution of $\kappa(X)$ on the calibration set, showing that the slope bound is finite and can be estimated from data. **(b)** Violation rate of the nested property as a function of $\lambda$. As predicted by Proposition 3.3, increasing $\lambda$ beyond the empirical scale of $\kappa(X)$ drives the violation rate to 0.

(3)). See Appendix D for implementation details. We evaluate the sensitivity of ITCR to the choice of the graph-level factuality uncertainty function by replacing the MLP with alternative mappings (random forest and SVM). As shown in Table 2, all variants satisfy the target coverage guarantees under both no-false and no-miss objectives, confirming that coverage validity is model-agnostic with respect to the uncertainty function class. Efficiency differs across mappings, with the MLP-based ITCR achieving the highest efficiency. This suggests that modeling capacity primarily affects efficiency rather than coverage validity.

*Table 2.* **Sensitivity analysis of the function class for the graph-level factuality uncertainty function (defined in Eq** (3)**)** at $\alpha = 0.1$ on QA. Empirical coverage and efficiency are reported under both no-false and no-miss targets for different choices of the mapping classes, demonstrating that ITCR achieves valid coverage consistently across function classes.

| Method | No-false | | No-miss | |
|---|---|---|---|---|
| | Cov. | Eff.(%) ↑ | Cov. | Eff.(%) ↑ |
| ITCR-rf | ✓ 0.911 | 16.45 | ✓ 0.900 | 10.20 |
| ITCR-svm | ✓ 0.893 | 18.71 | ✓ 0.905 | 6.85 |
| ITCR | ✓ 0.899 | 25.34 | ✓ 0.900 | 10.29 |

**Results: The nestedness condition in Proposition 3.3 is attainable in practice.** We assess the applicability of Proposition 3.3 by estimating $\kappa(X)$ from the calibration set. Fig. 4(a) shows that the values are finite, indicating that the slope bound $\kappa$ in Proposition 3.3 is estimable in practice. We then validate the implication of the proposition by examining the violation rate of the nested property as a function of $\lambda$. Fig. 4(b) shows that the violation rate decreases monotonically and vanishes once $\lambda$ exceeds the empirical scale of $\kappa(X)$.

### 4.3. Downstream Reasoning Performance

**Evaluation Protocol: LLMs.** We evaluate downstream reasoning performance across multiple LLMs with different model sizes and reasoning configurations. Specifi-

cally, we consider three models: (i) LLaMA-3.1-8B-Instruct (Grattafiori et al., 2024); (ii) Qwen3-4B-Thinking-2507 (Yang et al., 2025); and (iii) DeepSeek-R1-Distill-Qwen-1.5B (Guo et al., 2025). For a fair comparison, all models are evaluated on the GSM8K benchmark under the same experimental protocol. We fix the target confidence level at 80% and evaluate on 500 randomly sampled test questions for LLaMA, and on a randomly selected subset of 50 questions for Qwen and DeepSeek. All decoding and reasoning configurations follow the default settings of each backbone.

*Table 3.* **Comparison of ITCR and PostCal across different LLM backbones** on GSM8K and MATH datasets. See Appendix C for implementation details. All metrics are reported in percentage (%), and the better result in each pair is highlighted in **bold**. ITCR consistently yields larger net gains (PCR–NCR) over baselines by 18.77% on average, across both benchmarks.

| Dataset | Metric | LLaMA-8B | | Qwen3-4B | | DS-1.5B | |
|---|---|---|---|---|---|---|---|
| | | PostCal | ITCR | PostCal | ITCR | PostCal | ITCR |
| GSM8K | PCR↑ | 12.35 | **43.21** | 33.33 | **37.04** | 11.76 | **58.82** |
| | NCR↓ | 70.43 | **16.34** | 39.13 | **17.39** | 75.76 | **45.45** |
| | PCR–NCR↑ | -58.08 | **26.87** | -5.80 | **19.65** | -64.00 | **13.37** |
| MATH | PCR↑ | 8.33 | **20.83** | 4.88 | **43.90** | 8.33 | **16.67** |
| | NCR↓ | 76.92 | **3.85** | 44.44 | **22.22** | 50.00 | **2.63** |
| | PCR–NCR↑ | -68.59 | **16.98** | -39.56 | **21.68** | -41.67 | **14.04** |

**Evaluation Protocol: LLM Metrics.** We evaluate answer-level correction performance using conditional metrics defined over the original model answer, the conformalized answer, and a reference answer obtained via randomized rethinking. Let $a'$ be the conformalized answer and $a_\mathcal{M}$ denote the original model one, where $a_\mathcal{M} = 0$ and $a_\mathcal{M} = 1$ means incorrect or correct, respectively. Answer correctness is evaluated against a reference answer $a_{\mathrm{ref}}$, following prior work (Creswell et al., 2023; Wang et al., 2023b; Lightman et al., 2024). We report the following correction metrics.

(i) The **positive correction ratio (PCR)** measures the fraction of originally incorrect answers that are corrected after conformalization, defined as $\mathrm{PCR} := \mathbb{E}[\mathrm{Acc}(a_{\mathrm{ref}}, a') \mid \mathrm{Acc}(a_{\mathrm{ref}}, a_\mathcal{M}) = 0]$.

(ii) The **negative correction ratio (NCR)** measures the fraction of originally correct answers that become incorrect after conformalization, defined as $\mathrm{NCR} := \mathbb{E}[1 - \mathrm{Acc}(a_{\mathrm{ref}}, a') \mid \mathrm{Acc}(a_{\mathrm{ref}}, a_\mathcal{M}) = 1]$.

(iii) The effect of applying CP on LLM reasoning is summarized by **correction gain**, defined as $\mathrm{PCR} - \mathrm{NCR}$.

**Experimental Design.** We evaluate downstream answer correction performance under an inference-time control setting, where the model may invoke additional reasoning steps to deliberately refine its solution (Wu et al., 2025). The original answer $a_\mathcal{M}$ is produced using a thinking policy that stochastically invokes refine-thinking steps with a fixed probability. We compare two conformalized strategies: (i) PostCal (Rubin-Toles et al., 2025), which performs post-hoc

CP after a full reasoning trajectory is generated, and (ii) ITCR, which integrates CP into inference-time reasoning.

**Result: Inference-time conformal reasoning consistently improves downstream answer correction across LLM backbones.** As shown in Table 3, ITCR consistently achieves positive correction gain (PCR–NCR) across all LLM backbones with average 18.77% improvements, whereas PostCal exhibits negative or substantially lower gains. This improvement can be attributed to the ability of inference-time calibration to identify high-risk reasoning steps early and focus re-thinking on these steps, thereby avoiding unnecessary or harmful reasoning and improving the effectiveness of self-correction.

*Table 4.* **Average token usage and runtime comparison** between ITCR and PostCal across different LLM backbones on GSM8K. ITCR incurs lower token consumption and inference time.

| LLM Backbone | Method | Avg. Token↓ | Avg. Runtime (s)↓ |
|---|---|---|---|
| LLaMA-3.1-8B-Instruct | PostCal | 2092.80 | 67.69 |
| | **ITCR** | **1919.50** | **62.40** |
| Qwen3-4B-Thinking-2507 | PostCal | 1975.56 | 64.82 |
| | **ITCR** | **1812.92** | **55.04** |
| DeepSeek-R1-Distill-Qwen-1.5B | PostCal | 2578.42 | 87.07 |
| | **ITCR** | **2559.58** | **83.94** |

**Result: ITCR uses fewer tokens and has lower runtime than PostCal across all backbones.** Table 4 reports end-to-end computational cost measurements on GSM8K dataset. Both ITCR and PostCal share the same graph-construction and scoring pipeline; they differ only in whether conformal control is applied at inference time or post hoc. We observe that ITCR is cheaper than the post-hoc alternative, indicating that its correction gains do not come at the expense of higher computational cost.

*Table 5.* **Ablation study on the no-false and no-miss objectives** in ITCR with LLaMA-3.1-8B-Instruct. We evaluated the 59/500 GSM8K cases whose correct prefix contains ≤ 3 nodes.

| Dataset | Method | PCR↑ | NCR↓ | PCR–NCR↑ |
|---|---|---|---|---|
| GSM8K | No-false | **30.00** | 27.59 | 2.41 |
| | No-miss | 26.67 | **20.69** | **5.98** |

**Result: No-miss objective achieves a reasonable trade-off against over-truncation.** The output of the no-miss objective is intentionally recall-conservative: it may retain a few incorrect nodes to avoid removing factually correct steps needed downstream. At $\alpha = 0.1$, it retains about 0.5 incorrect nodes per graph on average (0.54/0.50/0.56 on GSM8K/MATH/QA), with incorrect-node fractions of 11.7%/9.1%/25.4%. Thus, the retained noise is limited in practice and reflects the intended trade-off against over-truncation. As shown in Table 5, the no-miss objective achieves a higher correction gain (PCR–NCR) on these heavily truncated cases. This supports its intended role: when

errors occur early, tolerating limited residual uncertainty helps preserve downstream performance. See additional experimental results and analysis in Appendix E.

## 5. Related Work

**Conformal Prediction.** Conformal Prediction (CP) is a powerful uncertainty quantification (UQ) framework initially proposed in (Vovk et al., 2005), which provides prediction sets for variables of interest with statistical guarantees. The main idea of CP is to substitute a model's point estimates with prediction sets for quantifying model uncertainty (Toccaceli, 2022; Vovk, 2014). Recent studies have introduced various conformity measures for both regression tasks (Papadopoulos et al., 2008; 2011) and classification tasks (Maltoudoglou et al., 2022). Compared to other CP frameworks, such as cross-validation (Vovk, 2015) or jackknife (Barber et al., 2021), many methods adopt the split approach (Papadopoulos et al., 2002; Lei et al., 2015), which enables the development of faster and more scalable CP algorithms. However, CP usually suffers inefficiency when there are no well-calibrated probabilities, with the intuition that larger prediction sets cover higher uncertainty. Much of the recent focus in the CP community has been on achieving desirable efficiency beyond validity (Sadinle et al., 2019; Romano et al., 2020; Shi et al., 2024; Wang et al., 2025). In addition to foundational works, CP techniques were previously extended and applied in the language domain, which are expected to not only provide accurate answers but also to let NLP systems "know when they do not know" (Campos et al., 2024). Despite the widespread use of CP methods in several NLP tasks such as text classification (Zhan et al., 2022; Giovannotti, 2022) and sequence tagging (Dey et al., 2022), there is little work on their application to language generation tasks.

**Conformal Language Model.** Several works have explored conformal language modeling (CLM) in settings with bounded or discretized output spaces, including question answering (Li et al., 2024; Zhang et al., 2020; Kumar et al., 2023; Rouzrokh et al., 2024) and token-level prediction (Ravfogel et al., 2023; Ulmer et al., 2024). Extending CLM to large language models (LLMs) with genuinely unbounded output spaces remains challenging, as classical conformal methods implicitly require enumerating or approximating the output space. To address this, (Wang et al., 2024b) selects prediction sets by grouping generations with similar uncertainty, but its effectiveness depends on strong alignment between nonconformity scores and correctness criteria, while (Quach et al., 2024; Shahrokhi et al., 2025) calibrates a stochastic sampling process to generate acceptable responses from the infinite output space. To obtain conformal factual guarantees for open-ended generation, (Mohri & Hashimoto, 2024) decomposes outputs

into claims and applies post-hoc filtering to remove low-confidence claims, leveraging claim-level fact verification (Min et al., 2023). Subsequent work improves information retention by rewriting underspecified claims (Jiang et al., 2025) or introducing conditional guarantees (Cherian et al., 2024). However, these approaches operate primarily at the claim level and can be brittle in reasoning domains, where correctness depends on the validity of preceding steps and locally correct claims may still be invalid. To address this, (Rubin-Toles et al., 2025) enforces coherent factuality by applying conformal prediction to reasoning graphs and evaluating entire claim orderings. Nevertheless, their method relies on post-hoc calibration and pruning over fully generated reasoning graphs, preventing inference-time generation with risk control and making it sensitive to hallucination-induced noise, which can lead to unstable thresholds and reduced efficiency.

## 6. Limitations and Future Directions

ITCR inherits the standard exchangeability assumption of split conformal prediction. When the calibration and test distributions differ, for example, due to changes in task domains, prompts, model backbones, or retrieval environments, the calibrated threshold may no longer achieve the target coverage. Extending ITCR to such non-exchangeable settings through localized calibration (Guan, 2023), importance reweighting (Barber et al., 2023), or online recalibration is an important future direction.

Another limitation is that the guarantee is defined with respect to the constructed reasoning graph and its factuality labels. Imperfect claim decomposition, dependency extraction, or factuality verification may affect how well the calibrated guarantee reflects semantic factuality in the original reasoning trace. Future work could account for uncertainty in graph construction and develop more adaptive nestedness penalties to improve the efficiency–robustness trade-off during inference-time stopping.

## 7. Conclusion

In this paper, we proposed *Inference-Time Conformal Reasoning (ITCR)*, a framework for calibrated factuality control during structured LLM reasoning. ITCR integrates conformal calibration into the generation process itself, using graph-level factuality uncertainty and a nested nonconformity score to decide when to stop expanding a reasoning graph. This yields valid coverage guarantees for structural factuality under no-false and no-miss objectives. Experiments across multiple datasets and LLM backbones show that ITCR achieves empirically valid coverage, improves the coverage–efficiency trade-off, and produces more accurate downstream reasoning by $18.77\%$ on average.

## Impact Statement

This paper proposes a statistical framework for inference-time uncertainty control in structured LLM reasoning. Its potential positive impact lies in improving the reliability and safety of multi-step reasoning by providing calibrated factuality guarantees. As with other advances in large language models, the same techniques could be misused or over-relied upon in sensitive settings, and responsible deployment with appropriate human oversight remains essential. The work is methodological in nature and does not target any specific application domain.

## Acknowledgments

Huan Zhang is supported in part by the AI2050 program at Schmidt Sciences (AI2050 Early Career Fellowship). Yan Yan is supported by the USDA-NIFA funded AgAID Institute award 2021-67021-35344, and the NSF grant CNS-2312125, IIS-2443828, DUE-2519063. The views expressed are those of the authors and do not reflect the official policy or position of the USDA-NIFA and NSF.

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

# A. Technical Proofs

In this section, we prove Proposition 3.3, Theorem 3.5 and 3.7.

## A.1. Proof of Proposition 3.3

*Proposition (Re) 1.* (Proposition 3.3 restated, condition for nested monotonicity). Let $B(U) = 1 - \sigma(\text{FU}^\theta(U, \{\text{fu}(v)\}_{v \in V_U}))$. For each input $X \sim P_X$, consider all ancestor-closed expansions $U \subset U' \subseteq G_X$, and define

$$\kappa := \sup_{X \sim P_X} \sup_{U \subset U' \subseteq G_X} \frac{\big(B(U) - B(U')\big)^+}{|V_{U'}| - |V_U|}.$$

If $\lambda \geq \kappa$, then the non-conformity score $S(U)$ satisfies the nested property.

*Proof.* (of Proposition 3.3)

Recall $S(U) = B(U) + \lambda|V_U|$. Then

$$\begin{aligned}
S(U') - S(U) &= \big(B(U') + \lambda|V_{U'}|\big) - \big(B(U) + \lambda|V_U|\big) \\
&= \lambda\big(|V_{U'}| - |V_U|\big) - \big(B(U) - B(U')\big) \\
&\geq \lambda\big(|V_{U'}| - |V_U|\big) - \big(B(U) - B(U')\big)^+,
\end{aligned} \tag{5}$$

where the inequality follows from the fact that $-(a) \geq -(a)^+$ for any $a \in \mathbb{R}$.

Recall that $\kappa = \sup_{X \sim \mathcal{P}_X} \sup_{U \subset U' \subseteq G_X} \frac{\big(B(U) - B(U')\big)^+}{|V_{U'}| - |V_U|}$, thus $\big(B(U) - B(U')\big)^+ \leq \kappa\big(|V_{U'}| - |V_U|\big)$ for any $U \subset U' \subseteq G_X$, where $X \sim \mathcal{P}_X$.

Thus, we have:

$$\lambda\big(|V_{U'}| - |V_U|\big) - \big(B(U) - B(U')\big)^+ \geq \lambda\big(|V_{U'}| - |V_U|\big) - \kappa\big(|V_{U'}| - |V_U|\big) = \big(\lambda - \kappa\big)\big(|V_{U'}| - |V_U|\big). \tag{6}$$

Combining Eq (5) and (6), we have:

$$S(U') - S(U) \geq \lambda\big(|V_{U'}| - |V_U|\big) - \big(B(U) - B(U')\big)^+ \geq \big(\lambda - \kappa\big)\big(|V_{U'}| - |V_U|\big).$$

If $\lambda \geq \kappa$, then $S(U') \geq S(U)$ for any $U \subset U'$.

Applying this monotonicity to each adjacent pair $(U^t, U^{t+1})$ along any inference-time trajectory, yielding $S(U^1) \leq \cdots \leq S(U^T)$, which is Definition 3.1.

Thus, we finish the proof of Proposition 3.3.

$\square$

## A.2. Proof of Theorem 3.5

*Theorem (Re) 1.* (Theorem 3.5 restated, no-false coverage guarantee). Suppose $S(\cdot)$ satisfies the nested property. Let $\widehat{U}^{\text{nf}}$ denote the output of Algorithm 1 with threshold $\tau_\alpha^{\text{nf}}$. Then the inference procedure achieves no-false coverage at level $1 - \alpha$, i.e., $\mathbb{P}\big(V_{\widehat{U}^{\text{nf}}} \subseteq \mathcal{T}\big) \geq 1 - \alpha$.

*Proof.* (of Theorem 3.5)

For simplification, we use $\widehat{U}$ to replace $\widehat{U}^{\text{nf}}$ in this proof.

Then, we transfer the coverage $\mathbb{P}\big(V_{\widehat{U}} \subseteq \mathcal{T}\big) \geq 1 - \alpha$ in Theorem 3.5 into its equivalent false-positive-rate (FPR) form:

$$\mathbb{P}\big(V_{\widehat{U}} \subseteq \mathcal{T}\big) \geq 1 - \alpha \iff \mathbb{P}\big(V_{\widehat{U}} \nsubseteq \mathcal{T}\big) \leq \alpha. \tag{7}$$

Recall that $\{G_i\}_{i=1}^n$ be the calibration set. For each $G_i$, the fixed expansion rule produces a nested sequence of ancestor-closed subgraphs $U_i^0 \subset U_i^1 \subset \cdots \subset U_i^{T_{G_i}}$,

Define the earliest-bad index $t_i^\star := \min\{t : V_{U_i^t} \nsubseteq \mathcal{T}\}$ with the convention $t_i^\star = +\infty$ if no such index exists. Correspondingly, define the score $Z_i := S\left(U_i^{t_i^\star}\right)$ for each calibration sample with $t_i^\star < \infty$.

Let $\mathcal{I}_{\mathrm{bad}} := \{i : t_i^\star < \infty\}$ and $m := |\mathcal{I}_{\mathrm{bad}}|$. Define the threshold $\tau_\alpha$ as the $\lfloor \alpha(m+1) \rfloor$-th largest value of $\{Z_i\}_{i \in \mathcal{I}_{\mathrm{bad}}}$.

Next, we analyze the output of Algorithm 1 during test-time. For a test input $X_{n+1}$, let $U_{n+1}^0 \subset U_{n+1}^1 \subset \cdots \subset U_{n+1}^{T_{G_{n+1}}}$ be the nested subgraphs generated by the same expansion rule. Algorithm 1 returns $\widehat{U}_{n+1} := U_{n+1}^{\hat{t}}$, where $\hat{t} := \max\{t : S(U_{n+1}^t) \le \tau_\alpha\}$.

Assume that for any input $G$, along its expansion trajectory $U_G^0 \subset U_G^1 \subset \cdots \subset U_G^{T_G}$, the nonconformity score satisfies $S(U_G^0) \le S(U_G^1) \le \cdots \le S(U_G^{T_G})$.

Define $t^\star := \min\{t : V_{U_{n+1}^t} \nsubseteq \mathcal{T}\}$ with $t^\star = +\infty$ if no such index exists. Since $U_{n+1}^t \subset U_{n+1}^{t'}$ for $t \le t'$, we have $V_{U_{n+1}^t} \subseteq V_{U_{n+1}^{t'}}$, hence $\{V_{U_{n+1}^t} \nsubseteq \mathcal{T}\} \subseteq \{V_{U_{n+1}^{t'}} \nsubseteq \mathcal{T}\}$. Therefore, if $V_{U_{n+1}^{\hat{t}}} \nsubseteq \mathcal{T}$, then $\hat{t} \ge t^\star$.

On the failure event $V_{\widehat{U}} \nsubseteq \mathcal{T}$, we have $t^\star \le \hat{t}$. By the definition of $\hat{t}$, $S(U_{\hat{t}}) \le \tau_\alpha$. By monotonicity and $t^\star \le \hat{t}$, it follows that

$$S(U_{t^\star}) \le S(U_{\hat{t}}) \le \tau_\alpha.$$

Therefore, we have:

$$\{V_{\widehat{U}} \nsubseteq \mathcal{T}\} \subseteq \{t^\star < \infty\} \cap \{S(U_{t^\star}) \le \tau_\alpha\}.$$

Define the test score $Z_{m+1} := S(U_{t^\star})$. By exchangeability of $(G_1, \ldots, G_n, G_{n+1})$ and the identical construction of $\{Z_i\}$ and $Z_{m+1}$, the multiset $\{Z_i\}_{i \in \mathcal{I}_{\mathrm{bad}}} \cup \{Z_{m+1}\}$ is exchangeable.

The split-conformal quantile construction then yields (Vovk et al., 2005; Shafer & Vovk, 2008; Angelopoulos & Bates, 2022)

$$\mathbb{P}(Z_{m+1} \le \tau_\alpha) \le \alpha.$$

Combining with the previous inclusion gives

$$\mathbb{P}(V_{\widehat{U}} \nsubseteq \mathcal{T}) \le \alpha,$$

According to Eq (7), we have:

$$\mathbb{P}(V_{\widehat{U}} \subseteq \mathcal{T}) \ge 1 - \alpha,$$

which completes the proof of Theorem 3.5.

$\square$

## A.3. Proof of Theorem 3.7

*Theorem (Re) 2.* (Theorem 3.7 restated, no-miss coverage guarantee). Suppose that $S(\cdot)$ satisfies the nested property. Let $\widehat{U}^{\mathrm{nm}}$ be the output returned by Algorithm 1 with threshold $\tau_{1-\alpha}^{\mathrm{nm}}$. Then the inference procedure satisfies no-miss coverage at level $1 - \alpha$, i.e., $\mathbb{P}\left(\mathcal{T} \subseteq V_{\widehat{U}^{\mathrm{nm}}}\right) \ge 1 - \alpha$.

*Proof.* (of Theorem 3.7)

The argument follows the same conformal calibration template as in the proof of Theorem 3.5 in Appendix A.2, with the coverage event adapted to the no-miss objective. Similarly, we use $\widehat{U}$ to replace $\widehat{U}^{\mathrm{nm}}$ in this proof for simplification.

Recall that no-miss coverage requires

$$\mathbb{P}(\mathcal{T} \subseteq V_{\widehat{U}}) \ge 1 - \alpha,$$

i.e., all true nodes are contained in the predicted ancestor-closed subgraph with probability at least $1 - \alpha$. Equivalently, a violation occurs if the generated subgraph misses at least one true node.

We then apply the same rank-based quantile argument on the calibration set, now using the $(1 - \alpha)$-quantile of the nonconformity scores over minimal ancestor-closed supergraphs of the true subgraph. Monotonicity of the score ensures that the stopping time $\widehat{U}$ is the minimal expansion whose score does not exceed the calibrated threshold.

The remaining steps follow verbatim from the proof of Theorem 3.5 in Appendix A.2 and are therefore omitted.

$\square$

## B. More Details on Datesets

**Frequency score used for risk estimation.** We use *frequency score* computed by sampling $n_{\text{samples}}$=5 continuations conditioned on the original question and each step's subclaim following (Mohri & Hashimoto, 2024). The resulting per-node scores are then aggregated by the learned graph-level uncertainty function for graph-level risk estimation.

**Reasoning graph construction.** Given a question and the list of generated subclaims, we construct a directed dependency graph following (Rubin-Toles et al., 2025), where each subclaim is a node and edges represent logical dependencies. The graph is represented as an $N \times N$ adjacency list (matrix form), where an edge $i \to j$ indicates that subclaim $j$ depends on subclaim $i$. The prompt we use to generate adjacency lists are listed following:

```
You are a system designed to create dependency graphs for subclaims in response to a given
↪  question.
Your output must strictly adhere to the following instructions:

1. Graph Description:
- Represent dependency relationships between subclaims as a directed graph.
- Each subclaim is a vertex in the graph.
- An edge (b -> a) exists if subclaim "a" depends on subclaim "b".
- Subclaims that are a priori (e.g., assumptions or definitions) should not have any
↪  ancestors.

2. Output Format:
- Provide your graph as an adjacency list of size NUM x NUM, where NUM is the number of
↪  subclaims.
- A template adjacency list with all entries zero will be included at the end of the
↪  prompt as reference.
- Replace 0s with 1s where relevant dependencies exist.
- A value of 1 at position i in row j indicates that subclaim j depends on subclaim i.
- A value of 0 indicates no dependency.
- Ensure no claim depends on itself (diagonal entries must be 0).

3. Rules:
- The adjacency list must be square with exactly NUM rows and NUM columns.
- Each row must contain exactly NUM integers.
- Output must consist solely of the adjacency list (e.g., [[0,1,0],[0,0,1],[0,0,0]]).
- Do not include explanations, commentary, or any other formatting.

4. Dependencies:
- Consider explicit and implicit dependencies between subclaims.
- Always represent dependencies, even if the subclaims are incorrect.

Now provide your adjacency list for the following question and subclaims:
Question: <QUESTION TEXT>

NUM = <N>
Subclaims:
1. <subclaim 1>
2. <subclaim 2>
...
N. <subclaim N>

Template:
[[0,0,...,0],
 [0,0,...,0],
```

```
 ...
 [0,0,...,0]]
```

## C. Implementation of Thinking via Inference-Time Calibration

For reproducibility, we report the prompt templates used for (i) step generation, and (ii) Step intervene.

**Step generation.** We enforce a strict JSONL schema for step-by-step reasoning and a single final answer line.

```
[System]
Solve the problem step by step.
Output exactly ONE line per step using the strict JSON format below:
Step t: {"subclaim": "<one short reasoning step>", "gpt-score": <number>}
Here, gpt-score is your confidence that this step is correct.

Rules:
- Use only the two keys: subclaim and gpt-score.
- Do not output any extra text.
- Do not output 'Answer:' until all steps are finished.

Finally output exactly one line:
Answer: <final numeric answer>

[User]
Problem: {problem}
```

Although the model outputs a self-reported `gpt-score` during step generation, our pipeline replaces it with a *frequency score* and the resulting frequency score is used as the step-level risk signal for reasoning subgraph risk estimation. Given a question and the list of generated subclaims, we construct a directed dependency graph required by our conformal inference procedure.

**Step intervene.** When current reasoning subgraph is flagged as high risk, we rewrite only the current step while keeping previous steps fixed, and then continue generation from a fixed prefix. Concretely, we prompt the same backbone LLM with a dedicated system–user prompt pair:

```
[System]
Rewrite one math-solution step in strict JSON.
Output exactly one line in this format:
  Step t: {"subclaim": "<one short reasoning step>", "gpt-score": <number>}
Use only the keys subclaim and gpt-score.
gpt-score must be one of: 1, 0, -1.
Do not output explanations, markdown, or extra text.

[User]
Problem: {problem}

Step {t}: {current_step}

This step may be unreliable. Rewrite it more carefully.
Rewrite ONLY Step {t} to be correct and consistent with earlier steps.
Output exactly one line: Step {t}:
```

The rewritten step inherits the same JSONL schema as the generation stage, so the subsequent pipeline (frequency scoring, graph construction, and conformal testing) proceeds without modification.

## D. Implementation of the Uncertainty Estimator $\mathrm{FU}_\theta$

We randomly split the dataset into a mapping set, a calibration set, and a test set with a ratio of $30\%/35\%/35\%$. The mapping set is used to learn the uncertainty estimator. Our uncertainty estimator is parameterized as the composition of a graph representation function $\phi$ and a classifier $f$, i.e.,

$$\mathrm{FU}_\theta = f \circ \phi$$

Each intermediate subgraph $U$ is embedded into a representation vector via

$$\phi(U, \{\mathrm{fu}(v)\}_{v \in V_U}) \in \mathbb{R}^d$$

Specifically, we construct the normalized graph Laplacian of $U$ and extract the smallest $k$ non-trivial eigenvalues as spectral features. Additionally, we compute summary statistics over the node-level confidence score $\mathrm{fu}(v)$, including the mean, standard deviation, minimum, and maximum across nodes in $V_U$.

We parameterize $f$ as a multi-layer perceptron (MLP) classifier. Formally,

$$f(\phi(U, \{\mathrm{fu}(v)\}_{v \in V_U})) = \mathrm{Sigmoid}(\mathrm{MLP}_\theta(\phi(U, \{\mathrm{fu}(v)\}_{v \in V_U})))$$

During training, we employ a binary focal loss:

$$\mathcal{L}_{\mathrm{focal}} = -\alpha_t(1 - p_t)^\gamma \log(p_t)$$

where $p_t$ denotes the predicted probability assigned to the true class, and $\alpha$ is class-dependent weighting. Unless otherwise stated, We use focal loss with hyperparameters $\alpha = 0.25$ and $\gamma = 2.0$ in all experiments. The model is optimized using the Adam optimizer with a learning rate of 0.001 and train for 500 epochs. The hidden dimension is set to 64, and batch size is set to 32.

# E. Additional Experimental Results

### E.1. Coverage-efficiency trade-offs

Figures 5 and 6 report the coverage-efficiency trade-offs as the target coverage level $1 - \alpha$ varies from 0.75 to 0.95 with a range 0.05, on GSM8K and MATH, respectively. Across all target coverage levels, ITCR consistently satisfies both no-miss and no-false coverage guarantees, while its empirical coverage closely tracks the target coverage $1 - \alpha$. At the same time, ITCR achieves the highest efficiency among all valid methods, with efficiency degrading smoothly as target coverage increases, reflecting near-tight inference-time generation rather than conservative over-coverage.

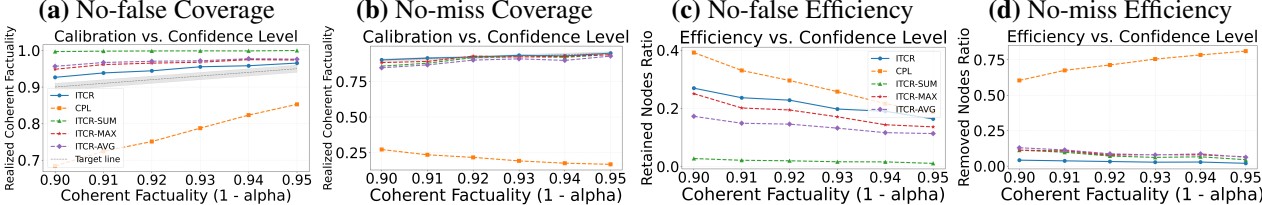

*Figure 5.* **Coverage and efficiency across target confidence levels** $1 - \alpha$ **under no-false and no-miss objectives** on GSM8K dataset. Subfigure **(a) and (b)** report the achieved empirical coverage as a function of the target coverage, with the dashed gray line indicating the ideal calibration line. Subfigure **(c) and (d)** report the corresponding efficiency, measured by the retained node ratio, where lower values indicate more compact reasoning subgraphs. ITCR remains close to the target calibration line and exhibits higher efficiency with valid coverage than baselines across target coverage levels.

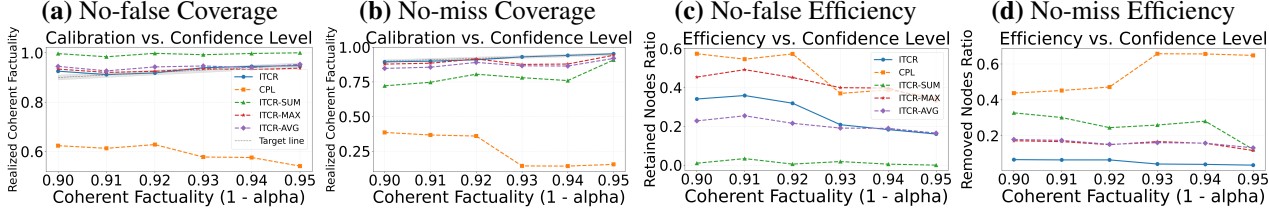

*Figure 6.* **Coverage and efficiency across target confidence levels** $1 - \alpha$ **under no-false and no-miss objectives** on MATH dataset. Subfigure **(a) and (b)** report the achieved empirical coverage as a function of the target coverage, with the dashed gray line indicating the ideal calibration line. Subfigure **(c) and (d)** report the corresponding efficiency, measured by the retained node ratio, where lower values indicate more compact reasoning subgraphs. ITCR remains close to the target calibration line and exhibits higher efficiency with valid coverage than baselines across target coverage levels.

## E.2. Sensitivity analysis on prompt variation

We conduct additional experiments with an alternative system prompt:

```
Solve the problem step by step.
Show your reasoning process and end with the final answer in the format:
Answer: \<final numeric answer\>
```

We set the target confidence level to 80% and evaluate on 50 randomly sampled test questions. As shown in Table 6, ITCR still substantially outperforms PostCal under the alternative CoT prompt across all three backbones, demonstrating that our method is robust to prompt variation.

*Table 6.* **Comparison of ITCR and PostCal across different LLM backbones** on GSM8K dataset under different prompt. All metrics are in %. The better result in each pair is in **bold**.

| LLM Backbone | Method | PCR↑ | NCR↓ | PCR–NCR↑ |
|---|---|---|---|---|
| LLaMA-3.1-8B-Instruct | PostCal | 6.06 | 70.59 | -64.53 |
| | **ITCR** | **21.21** | **5.88** | **15.33** |
| Qwen3-4B-Thinking-2507 | PostCal | 2.63 | 66.67 | -64.04 |
| | **ITCR** | **31.58** | **8.33** | **23.25** |
| DeepSeek-R1-Distill-Qwen-1.5B | PostCal | 17.14 | 33.33 | -16.19 |
| | **ITCR** | **48.57** | **26.67** | **21.90** |

*Table 7.* **Effect of dependency extraction on ITCR and PostCal performance** on GSM8K with LLaMA-3.1-8B-Instruct. All metrics are in %. The better result in each pair is in **bold**.

| Sample Type | Method | PCR↑ | NCR↓ | PCR–NCR↑ |
|---|---|---|---|---|
| Linear Chain (90%) | PostCal | 3.33 | 73.33 | -70.00 |
| | **ITCR** | **23.33** | **6.67** | **16.66** |
| All | PostCal | 6.06 | 70.59 | -64.53 |
| | **ITCR** | **21.21** | **5.88** | **15.33** |

Among 50 analyzed samples, 45/50 (90%) induce a purely linear dependency chain after step segmentation: each step depends only on its immediately preceding step, with no branching or skip dependencies. We evaluate ITCR and PostCal separately on these linear-chain samples, where the dependency structure is determined by the sequential output order and requires no additional extraction. The results in Table 7 show that ITCR still substantially outperforms PostCal on the linear-chain subset. Moreover, performance on this subset is nearly identical to that on the full set (PCR–NCR: 16.66 vs. 15.33), indicating that the correction gains of ITCR do not rely on nontrivial dependency extraction.

