# OpenReview forum: "Inference-Time Conformal Reasoning with Valid Factuality Control for Large Language Models"
_ICML.cc/2026/Conference — ICML 2026 regular_

### Official Review · Reviewer_iG7C · 2026-02-17

**Soundness:** 3
**Presentation:** 4
**Significance:** 3
**Originality:** 2
**Overall Recommendation:** 4
**Confidence:** 3

**Summary:**

This paper studies how to control factual errors during multi-step LLM reasoning, where intermediate steps depend on earlier steps and can be viewed as a directed acyclic graph (DAG) of claims rather than a flat list. The key observation is that in this setting, factuality is structural: if an early claim is wrong, many later claims that rely on it become unreliable, so uncertainty should be tracked over subgraphs, not just per-step scores. The authors propose Inference-Time Conformal Reasoning (ITCR), which integrates conformal prediction (CP) into the generation process itself instead of applying CP after the model finishes reasoning (post-hoc pruning). ITCR repeatedly expands a reasoning subgraph and uses a calibrated stopping rule: it stops generation once a graph-level “non-conformity” score exceeds a conformal threshold, and outputs the last accepted ancestor-closed subgraph.

**Compliance With Llm Reviewing Policy:**

Affirmed.

**Final Justification:**

concerns addressed

**Key Questions For Authors:**

1. How does ITCR behave under distribution shift between calibration and test (e.g., different domains, different prompting, different model checkpoints)? Since conformal validity relies on exchangeability, it would help to see stress tests or guidance for recalibration frequency.

2. No-miss objective allows “limited uncertainty when necessary.” In practice, how many incorrect nodes does the no-miss output tend to include, and how should users interpret that trade-off?

3.How are node-level factuality uncertainty scores obtained at inference time?

Addressing these points thoroughly would help my opinion of how robust the paper is.

**Limitations:**

Yes

**Strengths And Weaknesses:**

Strengths: Inference-time reliability control is a real practical gap: post-hoc pruning can waste compute and can produce invalid structures after pruning (ancestor closure repair). An inference-time stopping policy directly targets this issue. There is a clear formalization of structured reasoning as ancestor-closed subgraphs of a DAG and explicit definition of two coverage objectives (no-false and no-miss). Introducing and formalizing the no-miss objective as a recall-oriented complement to no-false is a useful perspective for structured reasoning where strict precision can cause degenerate abstention.

Weaknesses:
The approach relies on standard CP machinery plus a learned aggregator and a monotonicity penalty. The novelty is mainly in the integration and the nested-stopping framing, rather than a fundamentally new conformal method. The reported evaluation is limited to reasoning benchmarks with claim annotations and to settings where a reasoning graph and node-level factuality signals are available. The broader impact depends on how easily these prerequisites transfer to typical open-ended assistant settings.

---

> ### Author Rebuttal · Authors · 2026-03-31
>
> We thank the reviewer for the insightful review. We address each concern below with technical clarifications, **new experiments on the no-miss**, and details on the scoring protocol.
>
> **Q1: (1) The novelty is mainly in formulation; (2) The reported evaluation is limited**
>
> **A1:**
> (1) The reviewer is right that ITCR builds on standard CP machinery. However, the contribution is not integration but solving a specific open problem. Prior conformal LLM methods calibrate a fixed output post-hoc; inference-time stopping requires the score *repeatedly* during generation to be monotone under expansion, otherwise threshold crossing is reversible and stopping is invalid. No prior conformal score for LLMs provides this guarantee. Unlike Nested CP (Gupta et al., 2022), which assumes nestedness by construction, sequential generation requires nestedness to be enforced on top of a learned score; Proposition 3.3 gives the first sufficient condition for this. In addition, the no-miss objective is new: prior conformal LLM work targets precision-type guarantees and does not address the complete-abstention failure mode (Fig. 2a). We expand on this distinction further in our response to Reviewer 93z2 Q1.
>
> (2) Our current evaluation is limited to settings with explicit reasoning structure and claim-level factuality signals, a limitation shared by Mohri & Hashimoto (2024, ICML), Rubin-Toles et al.(2025, ICLR) and Cherian et al. (2024, NeurIPS). However, this limitation arises from upstream components (graph extraction and claim scoring), not ITCR's conformal stopping mechanism itself, which is modular and transfers directly as graph extraction and scoring improve (e.g., process reward models [r5] or automatic claim extraction [r6]).
>
> **Q2: How does ITCR behave under distribution shift?**
>
> **A2:**
> ITCR relies on split CP, which assumes calibration-test exchangeability — standard in split CP and shared by all baselines (i.e., CPL (Mohri & Hashimoto 2024, ICML) and PostCal (Rubin-Toles et al. 2025, ICLR)). Under shift, the finite-sample guarantee may therefore fail.
>
> Importantly, this limitation affects calibration, not the inference-time stopping mechanism itself. The graph-level score and stopping rule remain well-defined under shift; what may fail is the threshold transfer to the shifted test distribution. Hence, ITCR is modular with respect to shift-aware conformal corrections. Domain shift can be addressed via localized calibration [r7], while prompt and checkpoint changes are better suited to importance reweighting [r8]. Either correction applies at calibration time to provide valid coverage under distribution shift, without changing the core inference-time control algorithm. We will add this discussion in the revision.
>
> **Q3: (1) How many incorrect nodes in the no-miss output? (2) How should users interpret that trade-off?**
>
> **A3:**
> (1) The output of no-miss objective is intentionally recall-conservative: it may retain a few incorrect nodes to avoid removing factually correct steps needed downstream. At $\alpha=0.1$, it retains only about $0.5$ incorrect nodes per graph on average ($0.54/0.50/0.56$ on GSM8K/MATH/QA), with incorrect-node fractions of $11.7 \\%/9.1\\%/25.4 \\%$. Thus, the retained noise is limited in practice and reflects the intended trade-off against over-truncation.
>
> (2) This trade-off matters most when errors occur early. We evaluate the $59/500$ GSM8K cases whose correct prefix has $\le 3$ nodes, i.e., cases that deviate from the correct reasoning path almost immediately. On this subset, no-false and no-miss with LLaMA-3.1-8B-Instruct give:
>
> |Methods|PCR|NCR|PCR-NCR|
> |-|-|-|-|
> |no-false|30.00|27.59|2.41|
> |no-miss|26.67|20.69|5.98|
>
> Although no-miss retains slightly more uncertain content, it achieves higher correction gain (PCR−NCR) on these heavily truncated cases. This supports its intended role: when preserving reasoning context matters more than maximizing purity, allowing limited residual uncertainty can improve downstream correction.
>
> **Q4:  How to obtain node-level scores?**
>
> **A4:**
> Node-level scores are obtained online at inference time, following the frequency-based scoring protocol as Mohri & Hashimoto (2024, ICML), Rubin-Toles et al.(2025, ICLR) and Cherian et al. (2024, NeurIPS). Upon generating each subclaim, 5 continuations are sampled and their agreement yields a score in $[0,1]$. These per-node scores are then aggregated by the learned graph-level uncertainty function for the stopping rule. We will add a clearer description of this implementation to the paper.
>
> [r5] Lightman, H., et al. (2023). Let's verify step by step. ICLR
>
> [r6] Min, S., et al. (2023). Factscore: Fine-grained atomic evaluation of factual precision in long form text generation. EMNLP
>
> [r7] Guan, L. (2023). Localized conformal prediction: A generalized inference framework for conformal prediction. Biometrika
>
> [r8] Barber, R. F., et al. (2023). Conformal prediction beyond exchangeability." The Annals of Statistics

---

> > ### Author Rebuttal · Reviewer_iG7C · 2026-04-05
> >
> > Thanks for the reply, I will increase my score

---

### Official Review · Reviewer_tVAp · 2026-03-13

**Soundness:** 3
**Presentation:** 4
**Significance:** 3
**Originality:** 3
**Overall Recommendation:** 4
**Confidence:** 4

**Summary:**

The manuscript proposes inference-time conformal reasoning (ITCR), integrating CP into directed acyclic graph reasoning genration for structural factuality control. The porposed method outperforms post-hoc baselines.

**Compliance With Llm Reviewing Policy:**

Affirmed.

**Key Questions For Authors:**

see weaknesses

**Strengths And Weaknesses:**

+ nice motivation
+ strong empirical validity
+ downstream gains are also adequate

- experimental validation only limited to subset of reasoning benchmarks
- I think efficiency analysis is incomplete. Latency, tokens, runtime... analysis can be beneficial.

---

> ### Author Rebuttal · Authors · 2026-03-31
>
> We thank the reviewer for the insightful review. As suggested, we have provided **further benchmarks** and **efficiency analysis** to enable a more thorough evaluation of our method.
>
> **Q1: Experimental validation only limited to subset of reasoning benchmarks**
>
> **A1:**
> We agree that broader empirical coverage is valuable. At the same time, the current paper already goes beyond a single narrow setting: the experiments cover multiple reasoning benchmarks spanning mathematical reasoning and QA-style factual reasoning, and the downstream evaluation also includes three different LLM backbones rather than a single model family. Thus, while the benchmark scope can certainly be expanded, the current results are not confined to only one benchmark or one architecture.
>
> We also note that our setting requires structured claim-level factuality control over reasoning graphs, which substantially narrows the pool of usable benchmarks relative to standard final-answer reasoning evaluations. This is a shared constraint of this research direction (i.e., CPL (Mohri & Hashimoto 2024, ICML) and PostCal (Rubin-Toles et al. 2025, ICLR)) rather than something specific to our paper.
>
> To further address this concern, we additionally evaluated on the MATH benchmark, where all metrics are reported in percentage (\%):
>
> |Backbones|Methods|PCR|NCR|PCR-NCR|
> |-|-|-|-|-|
> |LLaMA-3.1-8B-Instruct|PostCal|8.33|76.92|-68.59|
> ||**ITCR**|20.83|3.85|16.98|
> |Qwen3-4B-Thinking-2507|PostCal|4.88|44.44|-39.56|
> ||**ITCR**|43.90|22.22|21.68|
> |DeepSeek-R1-Distill-Qwen-1.5B|PostCal|8.33|50.00|-41.67|
> ||**ITCR**|16.67|2.63|14.04|
>
> As shown, ITCR improves correction gains (PCR–NCR) by 17.57% on average. This additional result strengthens the claim that ITCR’s downstream advantage is not confined to a single benchmark.
>
> **Q2: Latency, tokens, runtime... analysis can be beneficial.**
>
> **A2:**
> We agree that end-to-end computational efficiency and runtime are also important metrics, and we therefore added a comparison of token cost and runtime on GSM8K. The table below reports the average token cost and average runtime per example for PostCal and ITCR.
>
> |Backbones|Methods|Avg. token cost|Avg. runtime (s)|
> |-|-|-|-|
> |LLaMA-3.1-8B-Instruct|PostCal|2092.80|67.69|
> ||**ITCR**|1919.50|62.40|
> |Qwen3-4B-Thinking-2507|PostCal|1975.56|64.82|
> ||**ITCR**|1812.92|55.04|
> |DeepSeek-R1-Distill-Qwen-1.5B|PostCal|2578.42|87.07|
> ||**ITCR**|2559.58|83.94|
>
> Compared with PostCal, ITCR uses fewer tokens and has lower runtime across all backbones, while also achieving stronger downstream correction performance (Table 3). This suggests that the correction gains of ITCR are not obtained at increased end-to-end computational cost; instead, ITCR provides a better utility-cost trade-off than the closest post-hoc baseline. We will include this comparison in the revised version.

---

> > ### Author Rebuttal · Reviewer_tVAp · 2026-04-02
> >
> > Authors have addressed my questions

---

### Official Review · Reviewer_cK9s · 2026-03-14

**Soundness:** 2
**Presentation:** 2
**Significance:** 2
**Originality:** 2
**Overall Recommendation:** 3
**Confidence:** 4

**Summary:**

The paper proposes an inference-time conformal method for controlling uncertainty over multi-step LLM reasoning. Instead of treating reasoning steps independently, it represents them as dependency-aware subgraphs and calibrates a stopping rule so the returned partial reasoning graph satisfies coverage goals such as "no false" or "no missed" steps.

**Compliance With Llm Reviewing Policy:**

Affirmed.

**Final Justification:**

I view conformal prediction for reasoning as a potentially interesting direction, but in this paper the guarantee is attached to an induced reasoning graph obtained through rigid decomposition and post hoc dependency extraction. The rebuttal improves the empirical support, but it does not change my core concern that the practical object being controlled is a constructed proxy rather than native model reasoning - i.e., this is not how models are trained, and the selected baseline models compose a quite random set without justification. For this reason, I am keeping my score unchanged.

**Key Questions For Authors:**

n/a

**Limitations:**

yes

**Strengths And Weaknesses:**

**Strengths**:

- The paper studies an important problem: controlling uncertainty over intermediate reasoning rather than only final answers. This is a meaningful direction for multi-step reasoning, where early mistakes can propagate to later steps.
- The paper treats reasoning as dependency-aware subgraphs and applies conformal calibration at inference time, which is a sensible extension beyond claim-level uncertainty control.
- The technical presentation is fairly clear. The paper states the monotonicity requirement, defines the score construction explicitly, and provides conformal guarantees under its assumptions.

**Weaknesses**:

- Although the paper is framed as inference-time control over "reasoning graphs," the practical setup depends on heavily engineered prompting that forces the model into a rigid stepwise subclaim format. Modern LLMs are not naturally trained to reason in this way, so part of the observed behavior may reflect instruction-following under an unusual setup rather than the proposed conformal control itself. A stronger evaluation would compare against standard plain prompting on the same verifiable reasoning benchmarks.
- The empirical comparison is fairly narrow. In the main evaluation, the paper appears to compare primarily against a single external baseline, CPL, while the remaining comparisons are largely internal heuristic variants such as MAX/SUM/AVG aggregation. Since the paper is motivated against prior structured or post-hoc control approaches, stronger matched comparisons to those alternatives would be important.
- The paper's efficiency claims are structural rather than computational. Efficiency is measured using retained or removed reasoning nodes under coverage constraints, but the actual pipeline requires rigid prompting, repeated sampling for frequency scores, and a separate graph-construction stage. As a result, the paper does not convincingly establish end-to-end inference efficiency in runtime or token cost.

---

> ### Author Rebuttal · Authors · 2026-03-31
>
> We thank the reviewer for the insightful review. As suggested, we have provided **new plain-CoT experiments**,  **runtime/token measurements**, and a clarification of the baseline comparison.
>
> **Q1: (1) Unusual rigid stepwise subclaim format (2) A stronger evaluation would compare against standard plain prompting.**
>
> **A1:**
> We respectfully disagree with this premise that our structured reasoning format constitutes a "rigid" or "unusual" setup for modern LLMs. Decomposing LLM outputs into atomic subclaims for factuality evaluation is the established methodology in this field, adopted by FActScore [r1], FELM [r2] and CPL [Mohri & Hashimoto (2024, ICML)]. This is not a limitation specific to our method. Even so, our ITCR vs. PostCal comparison holds prompting constant, both methods use identical structured reasoning outputs, claim representations, and factuality signals, differing only in when conformal control is applied. The performance gap in Table 3 therefore directly isolates the effect of inference-time vs. post-hoc control.
>
> **(2).** We additionally ran an additional experiment under standard zero-shot CoT [r3] to address external validity. The corresponding prompt is: "Solve the problem step by step. Show your reasoning process and end with the final answer in the format: Answer: \<final numeric answer\>". The table below summarizes the results on GMS8K, where all metrics are in percentage (\%):
>
> |Backbones|Methods|PCR|NCR|PCR-NCR|
> |-|-|-|-|-|
> |LLaMA-3.1-8B-Instruct|PostCal|6.06|70.59|-64.53|
> ||**ITCR**|21.21|5.88|15.33|
> |Qwen3-4B-Thinking-2507|PostCal|2.63|66.67|-64.04|
> ||**ITCR**|31.58|8.33|23.25|
> |DeepSeek-R1-Distill-Qwen-1.5B|PostCal|17.14|33.33|-16.19|
> ||**ITCR**|48.57|26.67|21.90|
>
> As shown, ITCR still substantially outperforms PostCal under plain CoT across all three backbones, with $20.16\\%$ correction gains (PCR–NCR). This matches our original results (Table 3): ITCR remains effective under standard zero-shot CoT prompting.
>
> **Q2: The empirical comparison is fairly narrow.**
>
> **A2:**
> We respectfully note that our comparison meets or exceeds the baseline norm in this subfield. Conformal factuality control for LLMs is a very recent area; examining published work:
> - CPL (Mohri & Hashimoto, 2024, ICML) uses no external CP baseline;
> - Cherian et al. (2024, NeurIPS) uses **one external baseline** (CPL). It is a variant of CPL with a different conditional coverage target.
> - PostCal (Rubin-Toles et al, 2025, ICLR) uses **one external baseline** (CPL).
>
> ITCR includes two external baselines, CPL for inference-time control (Table 1) and PostCal for post-hoc control (Table 3), plus three controlled ablations (MAX/SUM/AVG), which is at or above the field standard.
>
> More specifically:
> 1. CPL is the only existing method performing inference-time conformal control. The MAX/SUM/AVG variants replace our learned graph-level score with heuristic aggregations, isolating the contribution of learned scoring.
>
> 2. PostCal is the most direct post-hoc comparator, sharing identical generations and factuality signals with ITCR and differing only in when control is applied. Thus, the gap isolates the effect of inference-time calibration.
>
> We will clarify the distinct roles of external baselines vs. ablations in the revision.
>
> **Q3: The paper's efficiency claims are structural rather than computational.**
>
> **A3:**
> We provide end-to-end computational cost measurements on GSM8K below, which show that ITCR is in fact cheaper than the post-hoc alternative:
>
> |Backbones|Methods|Avg. token|Avg. runtime (s)|
> |-|-|-|-|
> |LLaMA-3.1-8B-Instruct|PostCal|2092.80|67.69|
> ||**ITCR**|1919.50|62.40|
> |Qwen3-4B-Thinking-2507|PostCal|1975.56|64.82|
> ||**ITCR**|1812.92|55.04|
> |DeepSeek-R1-Distill-Qwen-1.5B|PostCal|2578.42|87.07|
> ||**ITCR**|2559.58|83.94|
>
> ITCR uses fewer tokens than PostCal and also has lower runtime across all backbones. Both ITCR and PostCal share the same graph-construction and scoring pipeline; the main difference is whether conformal control is applied (inference-time vs. post-hoc). This shows that the correction gains of ITCR do not come at the expense of higher end-to-end computational cost.
>
> In our paper, our efficiency claims concern coverage-constrained compactness of the predicted reasoning subgraph, not runtime or token cost. This follows the standard conformal prediction usage of efficiency [r4], where the goal is to make the prediction object as compact as possible subject to valid coverage, rather than to optimize wall-clock runtime.
>
> [r1] Min, S., et al. (2023). Factscore: Fine-grained atomic evaluation of factual precision in long form text generation. EMNLP
>
> [r2] Zhao, Yiran, et al. (2023). Felm: Benchmarking factuality evaluation of large language models. NeurIPS
>
> [r3] Kojima, Takeshi, et al.(2022). Large language models are zero-shot reasoners." NeurIPS
>
> [r4] Angelopoulos, A. N., & Bates, S. (2023). Conformal prediction: A gentle introduction. Foundations and Trends in Machine Learning

---

> > ### Author Rebuttal · Reviewer_cK9s · 2026-04-04
> >
> > Thank you for the detailed rebuttal. The additional plain-CoT and runtime/token measurements are useful and strengthen the empirical section. That said, my overall assessment remains unchanged, since my main concern is not only about missing experiments, but about the reliance on a structured decomposition and dependency-extraction pipeline. In my view, this makes it unclear how well the proposed conformal control would transfer to more natural reasoning behavior of modern LLMs.

---

> > > ### Author Response · Authors · 2026-04-07
> > >
> > > **FQ1: How well the proposed conformal control would transfer to more natural reasoning behavior of modern LLMs?**
> > >
> > > **FA1:**
> > > Thank you for recognizing that our additional plain-CoT experiments and runtime/token measurements have addressed the empirical concerns regarding effectiveness under standard prompting and computational efficiency.
> > >
> > > We understand that the remaining question is whether ITCR is practical for modern LLMs, given its reliance on a structured decomposition and dependency-extraction pipeline. We believe **the answer is yes:** (1) stepwise decomposition is already consistent with current reasoning-model practice rather than an artificial constraint, and (2) in the dominant case (90\% of plain CoT samples), the dependency structure is linear and requires no additional extraction, yet ITCR still delivers substantial gains.
> > >
> > > ​​First, we respectfully disagree with the premise that stepwise decomposition is unnatural for modern reasoning LLMs. Recent reasoning models such as OpenAI o1 [fr1], DeepSeek-R1 [fr2], and Qwen3-Thinking [fr3] are explicitly trained to generate step-by-step reasoning traces during inference. Step-level reasoning is also the standard interface used in process reward modeling and step-level verification [fr4, fr5, fr6, fr7]. The step-level representation used by ITCR is therefore not an artificial format unique to our method, but one that is consistent with current reasoning-model practice.
> > >
> > > Even setting aside the above trend, we directly probe the transferability concern with an additional controlled analysis under plain zero-shot CoT on GSM8K with LLaMA-3.1-8B-Instruct.
> > > Among 50 analyzed samples, 45/50 (90\%) induce a purely linear dependency chain after step segmentation: each step depends only on the immediately preceding step, with no branching or skip dependencies. In this regime, the dependency structure is determined by the sequential output order, requiring no additional extraction, which directly addresses the practical concern about reliance on dependency extraction. We then evaluated ITCR and PostCal separately on these linear-chain samples:
> > >
> > > |Sample Types|Methods|PCR|NCR|PCR-NCR|
> > > |-|-|-|-|-|
> > > |Linear Chain (90\%)|PostCal|3.33|73.33|-70.00|
> > > ||**ITCR**|23.33|6.67|16.66|
> > > |All|PostCal|6.06|70.59|-64.53|
> > > ||**ITCR**|21.21|5.88|15.33|
> > >
> > > ITCR still substantially outperforms PostCal on the linear-chain subset, where the dependency structure is determined by the output order. Moreover, performance on the linear-chain subset is nearly identical to that on the full set (PCR–NCR: 16.66 vs. 15.33), showing that the gain does not rely on nontrivial dependency extraction.
> > >
> > > In summary, these results confirm that ITCR is readily applicable to standard stepwise reasoning used by modern LLMs, without nontrivial dependency extraction, and its gains persist in this setting.
> > >
> > > [fr1] Jaech, Aaron, et al. (2024). Openai o1 system card. arXiv
> > >
> > > [fr2] Guo, Daya, et al. (2025). Deepseek-r1: Incentivizing reasoning capability in llms via reinforcement learning. arXiv
> > >
> > > [fr3] Yang, An, et al. (2025). Qwen3 technical report." arXiv
> > >
> > > [fr4] Lightman, H., et al. (2023). Let's verify step by step. ICLR
> > >
> > > [fr5] Uesato, Jonathan, et al. (2022). Solving math word problems with process-and outcome-based feedback. arXiv
> > >
> > > [fr6] Rajendhran, Rishanth, et al. (2025). VERIFASTSCORE: Speeding up long-form factuality evaluation." arXiv
> > >
> > > [fr7] Pronesti, Massimiliano, Anya Belz, and Yufang Hou.(2026). Beyond Outcome Verification: Verifiable Process Reward Models for Structured Reasoning. arXiv

---

### Official Review · Reviewer_93z2 · 2026-03-23

**Soundness:** 3
**Presentation:** 3
**Significance:** 3
**Originality:** 3
**Overall Recommendation:** 4
**Confidence:** 2

**Summary:**

The paper argues that factual errors in LLM reasoning are structural: in multi-step reasoning, later claims depend on earlier ones, so uncertainty should be assessed over the whole reasoning graph, not just individual steps. It introduces inference-time conformal reasoning, which uses conformal prediction during generation to estimate graph-level factuality uncertainty and decide when to stop reasoning, providing theoretical coverage guarantees and producing more accurate results than post-hoc pruning.

**Compliance With Llm Reviewing Policy:**

Affirmed.

**Final Justification:**

Great for the new experimental results, but for a clear "accept", I'd still expect more extensive empirical results. So, I stick to my "weak accept".

**Key Questions For Authors:**

No questions.

**Limitations:**

yes

**Strengths And Weaknesses:**

The paper is well-presented and organized, and it seems to be sound.

The paper is following the natural question whether we can move from post-hoc calibration to inference-time conformal control of factuality in LLM reasoning while retaining coverage guarantees. As such, it is perhaps not very original, but the experimental results over several reasoning benchmarks and LLM architectures indicate that ITCR maintains empirically valid coverage and yields an average reasoning accuracy gain of 19.96%. it would be good to see the approach on further benchmarks.

---

> ### Author Rebuttal · Authors · 2026-03-31
>
> We thank the reviewer for the insightful review. As suggested, we have provided **further benchmarks** to demonstrate the effectiveness of our method, and we also want to clarify our originality.
>
> **Q1: it is perhaps not very original.**
>
> **A1:**
> We agree that the question is natural, but the extension is not direct.
>
> From the CP perspective, ITCR moves conformal prediction from post-hoc calibration of fixed outputs to inference-time control of structured generation. From the LLM reasoning perspective, it replaces heuristic intervention on reasoning trajectories with a calibrated stopping rule with coverage guarantees.
>
> This shift is technically non-trivial. Prior work applies CP after a full reasoning graph is generated, whereas ITCR uses CP during sequential expansion to decide when to stop. For this to be valid, the non-conformity score must be monotone along the generation trajectory; otherwise threshold crossing is not a valid stopping event. ITCR therefore introduces a monotone graph-level score and a formal nestedness condition, both specific to inference-time control. It also moves from node-level to graph-level uncertainty, since factual validity in structured reasoning depends on ancestor closure rather than isolated claims.
>
> Thus, the contribution is not only asking a natural question, but showing how CP can govern reasoning generation itself, rather than only filter completed outputs.
>
> **Q2: it would be good to see the approach on further benchmarks.**
>
> **A2:**
> We agree that evaluation on additional benchmarks would further strengthen the paper. That said, the current experiments already cover several reasoning benchmarks and multiple LLM architectures, with consistent conclusions in both the coverage evaluation and the downstream correction setting.
>
> To further address this concern, we additionally evaluated on the MATH benchmark, where all metrics are reported in percentage (\%):
>
> |Backbones|Methods|PCR|NCR|PCR-NCR|
> |-|-|-|-|-|
> |LLaMA-3.1-8B-Instruct|PostCal|8.33|76.92|-68.59|
> ||**ITCR**|20.83|3.85|16.98|
> |Qwen3-4B-Thinking-2507|PostCal|4.88|44.44|-39.56|
> ||**ITCR**|43.90|22.22|21.68|
> |DeepSeek-R1-Distill-Qwen-1.5B|PostCal|8.33|50.00|-41.67|
> ||**ITCR**|16.67|2.63|14.04|
>
> As shown, ITCR improves correction gains (PCR–NCR) by 17.57% on average. This additional result strengthens the claim that ITCR’s downstream advantage is not confined to a single benchmark.

---

> > ### Author Rebuttal · Reviewer_93z2 · 2026-04-04
> >
> > Great for the new experimental results, but for a clear "accept", I'd still expect more extensive empirical results. So, I stick to my "weak accept".

---

### Decision · Program_Chairs · 2026-04-30

**Decision:**

Accept (regular)

**Comment:**

This papers's take is that reasoning is structured, so uncertainty should be assessed over the whole reasoning graph, not just individual steps. It introduces inference-time conformal reasoning, which uses conformal prediction during generation to estimate graph-level factuality uncertainty and decide when to stop reasoning, providing theoretical coverage guarantees and producing more accurate results than post-hoc pruning.. This is a sound paper. The reviewers have pointed out some suggestions for improvements - please consider them for the next version.